# Single-cell lineage tracing identifies hemogenic endothelial cells in the adult mouse bone marrow

Jing-Xin Feng[1]*, Mei-Ting Yang[2,3], Lili Li[1], Caiyi C Li[4], Ferenc Livak[4], Jack Chen[5], Yongmei Zhao[5], Dunrui Wang[1], Avinash Bhandoola[4], Naomi Taylor[6], Giovanna Tosato[1]*†

[1]Laboratory of Cellular Oncology, Center for Cancer Research, National Cancer Institute (NCI), National Institutes of Health, Bethesda, United States; [2]School of Life Sciences, Northeast Normal University, Changchun, China; [3]Experimental Immunology Branch, Center for Cancer Research (CCR), National Cancer Institute (NCI), National Institutes of Health, Bethesda, United States; [4]Laboratory of Genome Integrity, Center for Cancer Research, National Cancer Institute (NCI), National Institutes of Health, Bethesda, United States; [5]Sequencing Facility Bioinformatics Group, Bioinformatics and Computational Science Directorate, Frederick National Laboratory for Cancer Research, Frederick, United States; [6]Pediatric Oncology Branch, Center for Cancer Research (CCR), National Cancer Institute (NCI), National Institutes of Health, Bethesda, United States

*For correspondence:
jingxinfeng475@gmail.com (J-XF);
tosatog@mail.nih.gov (GT)

†Senior author and lead contact

## eLife Assessment

This manuscript by Feng et al. provides **valuable** evidence regarding the hematopoietic differentiation of bone marrow endothelial cells in the adult mouse. Overall, the authors have addressed our main concerns. **Solid** data now more strongly support long-term multi-lineage reconstitution of the adult hemogenic endothelial cells. However, critical data, especially regarding the endothelial cells' hematopoietic identity and functional capacity, remain insufficient, which limits the strength of the hemogenic claim, especially the assertion that these adult hemogenic ECs generate bona fide HSCs. Additional experiments would be necessary to fully rule out alternative explanations.

**Abstract** During mouse development, hematopoietic stem and progenitor cells (HSPCs) originate from hemogenic endothelial cells (ECs) through a process of endothelial-to-hematopoietic transition. These HSPCs are thought to fully sustain adult hematopoiesis. However, it remains unknown whether adult ECs retain hemogenic potential. Here, we used in vivo genetic lineage tracking at population and single-cell (sc) levels, scRNA sequencing, and bone marrow (BM) transplantation to detect hemogenic ECs in adult mice. We identify and characterize BM-resident, adult *Cdh5/VE-Cadherin*+ ECs that produce hematopoietic cell-progeny in vitro and in mice. These adult hemogenic ECs and their hematopoietic cell progeny give rise to hematopoietic cells following adoptive transfer into adult mice. Furthermore, blood cells generated from adult and developmental ECs comparably home to peripheral tissues, where they similarly contribute to inflammatory responses. Thus, our results identify previously unrecognized BM-derived adult hemogenic ECs that generate HSPC and functional mature blood cells.

## Introduction

During development, hemogenic endothelial cells (ECs) generate hematopoietic cells through a process of endothelial-to-hematopoietic transition (EHT) at geographically defined anatomical sites (*Jaffredo et al., 1998*; *Zovein et al., 2008*; *Bertrand et al., 2010*; *Boisset et al., 2010*; *Yokomizo and Dzierzak, 2010*; *Chen et al., 2011*; *Frame et al., 2016*; *Rhodes et al., 2008*; *Nakano et al., 2013*). At these locations, the hemogenic ECs represent a small fraction of all ECs (*Goldie et al., 2008*; *Marcelo et al., 2013*; *Hirschi, 2012*), and their competency to produce hematopoietic stem and progenitor cells (HSPCs) is temporally restricted to short developmental windows, and the hemogenic potential differs (*Yoshimoto et al., 2012*; *Lin et al., 2014*; *Iturri et al., 2021*; *Soares-da-Silva et al., 2021*; *de Bruijn et al., 2002*; *Hadland et al., 2015*). In the dorsal aorta, the hemogenic endothelium produces hematopoietic stem cells (HSCs) and other multipotent progenitors between E10.5 and E11.5 (*de Bruijn et al., 2002*; *Hadland et al., 2015*; *Patel et al., 2022*; *Ding et al., 2010*) that persist in the adult and are generally believed to sustain adult hematopoiesis throughout life.

Efforts to reliably generate ex vivo HSC from Cdh5-expressing ECs identified difficulties in reproducing the microenvironmental clues coming from the inductive niche cells (*Ding et al., 2010*; *Kobayashi et al., 2010*; *Butler et al., 2012*) and have generally relied on transcription factor-induced reprogramming of the ECs to drive hematopoietic specification (*Yzaguirre and Speck, 2016*; *Zhu et al., 2020*). The transcription factor RUNX1, which marks the hemogenic endothelium, can confer a hemogenic potential to embryonic ECs lacking such potential (*North et al., 1999*; *Chen et al., 2009*; *Eliades et al., 2016*; *Yzaguirre et al., 2018*). When transcription factors RUNX1, FOSB, GFI1, and SPI1 were co-expressed in adult murine ECs co-cultured with 'vascular niche ECs', EHT was induced in vitro producing HSC with long-term self-renewal capacity (*Lis et al., 2017*). A similar approach was used with human ECs enabling hematopoietic specification (*Sandler et al., 2014*).

An unresolved question is whether hemogenic ECs, thought to be mostly restricted to the early stages of mouse development (*Hirschi, 2012*; *Yzaguirre and Speck, 2016*), may persist in the adult mouse (*Zovein et al., 2008*; *Hirschi, 2012*; *Pelosi et al., 2012*; *Kilani et al., 2019*). Exploiting advances in cell lineage tracking and single-cell analyses (*Pei et al., 2020*; *Hernandez et al., 2022*), we report the identification of hemogenic ECs in the adult mouse bone marrow (BM) that produce functional hematopoietic progenitors and mature blood cells.

## Results

### Evidence that adult BM ECs generate hematopoietic cells

We assessed the hemogenic potential of ECs in adult mice using Cre-reporter-based lineage tracing. Since *Cdh5*, encoding vascular endothelial cadherin (VE-Cadherin), is selectively expressed by ECs, Cdh5-Cre$^{ERT2}$ recombinase activity can allow tracking the hematopoietic cell output from adult hemogenic ECs (*Gentek et al., 2018*; *Wang et al., 2013*). Therefore, we generated three Cdh5-based lineage-tracing models using inducible Cdh5-Cre$^{ERT2}$(PAC) (*Sörensen et al., 2009*; *Pitulescu et al., 2010*) and Cdh5-Cre$^{ERT2}$(BAC) (*Okabe et al., 2014*) mouse lines, in combination with the Cre-reporter lines ZsGreen and mTmG (*Figure 1—figure supplement 1A*). We then treated 8- to 12-week-old mice with three doses of tamoxifen (10 mg kg$^{-1}$, gavage) on consecutive days, and 4 weeks later we evaluated peripheral blood and BM (*Figure 1A*). Expectedly (*Wang et al., 2013*), most BM Endomucin$^+$ ECs were ZsGreen$^+$ in tamoxifen-treated Cdh5-Cre$^{ERT2}$(PAC)/ZsGreen and Cdh5-Cre$^{ERT2}$(BAC)/ZsGreen mice, and virtually no ZsGreen$^+$ ECs were present in Cre negative or peanut oil-treated controls (*Figure 1—figure supplement 1B, C*). By flow cytometry, >90% CD31$^+$VE-Cadherin$^+$ BM ECs were tracked by tamoxifen-induced fluorescence in the three mouse lines (*Figure 1B*; *Figure 1—figure supplement 1D*). A low-level tamoxifen-independent reporter fluorescence was also detected in BM ECs, which was low in the mTmG reporter line (*Figure 1B*), and higher in the ZsGreen reporter lines, previously attributed to 'basal' Cre$^{ERT2}$ activity (*Álvarez-Aznar et al., 2020*; *Figure 1—figure supplement 1D*).

To evaluate the hemogenic potential of adult BM ECs, we analyzed the expression of the hematopoietic marker CD45 in Cdh5-tracked cells. Notably, tracked CD45$^+$ hematopoietic cells, presumed progeny of VE-Cadherin$^+$ ECs, were detected by flow cytometry in BM and blood of mice from all three Cdh5-Cre$^{ERT2}$ mouse lines (*Figure 1C, D*; and *Figure 1—figure supplement 1E, F*); representative flow cytometry gating in *Figure 1—figure supplement 1G, H*. Moreover, confocal microscopy

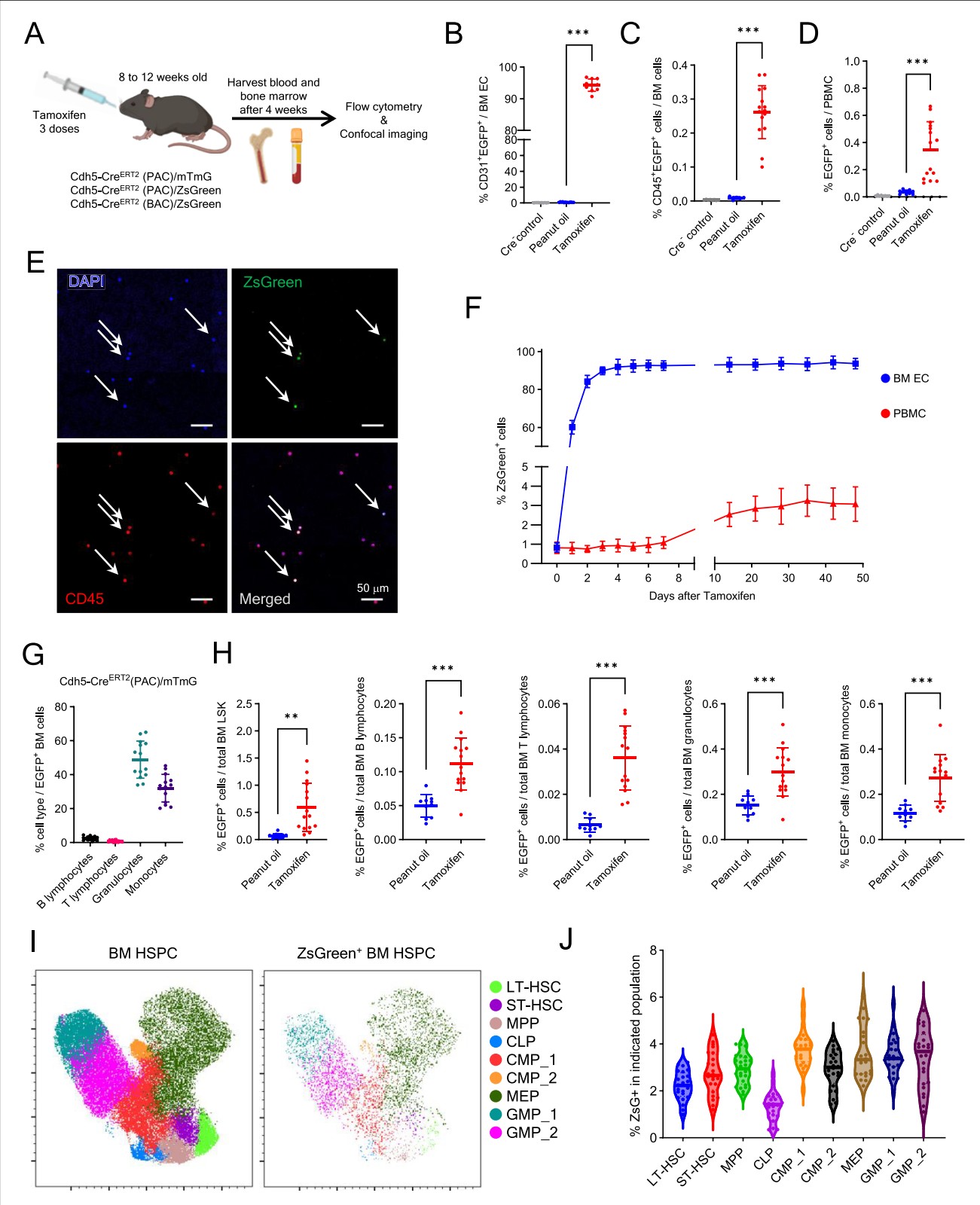

**Figure 1.** Lineage tracking discloses a contribution of endothelial cells (ECs) to hematopoiesis in adult bone marrow (BM). (**A**) Experimental design: tamoxifen was administered to 8- to 12-week-old Cdh5-Cre mice to induce fluorescent labeling of VE-Cadherin+ cells and their cell progeny. Four weeks later, BM and blood were analyzed. (**B**) CD31+EGFP+ BM ECs in Cre− mice (n = 10) and Cre+ mice treated with oil (n = 13) or tamoxifen (n = 10); flow cytometry results. (**C, D**) CD45+EGFP+ cells in BM and blood from Cre− mice (n = 8) and Cre+ mice treated with oil (n = 6–10) or tamoxifen (n =

*Figure 1 continued*

15–18). Representative flow cytometry gating in **Figure 1—figure supplement 1G**. (**E**) Representative blood smear from a tamoxifen-treated Cdh5-Cre[ERT2](PAC)/ZsGreen mouse showing ZsGreen$^+$CD45$^+$DAPI$^+$ cells (arrows). (**F**) Kinetics of ZsGreen$^+$ cell detection in BM ECs (CD45$^-$VE-Cadherin$^+$) and blood white blood cells (WBCs) post-tamoxifen; mouse n = 8–10/group. (**G**) EGFP$^+$ B and T-lymphocytes, granulocytes, and monocytes in BM of tamoxifen-treated mice (n = 14) as percent of total EGFP$^+$ cells; three experiments. (**H**) EGFP$^+$ BM LSK, lymphocytes, granulocytes, and monocytes as percent of total EGFP$^{+/-}$ cell type; Cdh5-Cre[ERT2](PAC)/mTmG mice (oil n = 10; tamoxifen n = 15), three experiments. (**I**) Uniform Manifold Approximation and Projection (UMAP) plots of Lin$^-$ BM hematopoietic stem and progenitor cell (HSPC) from tamoxifen-treated Cdh5-Cre[ERT2](PAC)/ZsGreen mice (n = 26; 1 femur/mouse) showing FlowSOM clustering of all (ZsGreen$^{+/-}$) and ZsGreen$^+$ populations. (**J**) Violin plots showing ZsGreen$^+$ cell distribution across HSPC subsets from (**I**). Dots represent individual mice; data shown as mean ± SD except shown as median in (**G**). *p < 0.05, **p < 0.01, ***p < 0.001 by Student's t-test.

The online version of this article includes the following figure supplement(s) for figure 1:

**Figure supplement 1.** Contribution of endothelial cells (ECs) to hematopoiesis in adult bone marrow (BM) is revealed by Cdh5-Cre[ERT2] mouse tracking lines.

**Figure supplement 2.** Characterization of tracked hematopoietic progenitors and mature cells in adult bone marrow (BM) and peripheral blood of Cdh5-Cre reporter mice.

identified isolated ZsGreen$^+$CD45$^+$ cells in BM (**Figure 1—figure supplement 1I**) and blood of tamoxifen-induced mice (**Figure 1E**). Importantly, while a small fraction of fluorescent CD45$^+$ cells were detected in EGFP and ZsGreen (**Figure 1C, D**, **Figure 1—figure supplement 1D–F**) reporter mice without tamoxifen, as reported (**Álvarez-Aznar et al., 2020**; **Stifter and Greter, 2020**; **Liu et al., 2010**; **Vooijs et al., 2001**), the significant increase of tamoxifen-induced fluorescent CD45$^+$ cells (**Figure 1C, D**, **Figure 1—figure supplement 1D–F**) indicates the occurrence of Cdh5-Cre recombination in the adult mouse, presumably tracking EHT. This tamoxifen-induced increase in CD45$^+$ZsGreen$^+$ peripheral blood mononuclear cells (PBMCs) was also observed in individual mice (**Figure 1—figure supplement 1J**).

We tested the kinetics of tamoxifen-induced ZsGreen expression in BM ECs and PBMC of Cdh5-Cre[ERT2](PAC)/ZsGreen mice (**Figure 1F**). By flow cytometry, virtually all BM ECs were ZsGreen$^+$ by day 4 after tamoxifen administration, and this level persisted over 50 days. Instead, the percentage of ZsGreen$^+$ PBMCs increased more gradually, plateauing around day 30, and this level persisted over ~50 days of observation. This gradual increase is likely attributable to tamoxifen-induced ZsGreen expression in hemogenic ECs.

Further analysis showed that most BM EGFP$^+$CD45$^+$ cells in Cdh5-Cre[ERT2](PAC)/mTmG mice were CD11b$^+$Ly6G$^+$ granulocytes (48.85%) and CD11b$^+$Ly6G$^-$ monocytes (32.06%), while CD19$^+$ B (2.46%) and CD3$^+$ T (0.84%) lymphocytes were detected at lower frequencies (**Figure 1G**). All EGFP$^+$ BM cell populations were significantly induced by tamoxifen administration, including rare LSK (Lin$^-$Sca1$^+$cKit$^+$) progenitors (**Figure 1H**). Also, the peripheral blood of tamoxifen-induced Cdh5-Cre[ERT2](PAC)/mTmG mice contained EGFP$^+$ granulocytes and monocytes, and fewer B and T lymphocytes (**Figure 1—figure supplement 1K, L**). Similarly, in ZsGreen-reporter mice, tracked LSK progenitors, B lymphocytes, granulocytes, and monocytes were all present in the BM and blood (**Figure 1—figure supplement 2A–D**).

We further characterized the tracked BM hematopoietic LSK stem/progenitors by flow cytometry (**Figure 1—figure supplement 2E**). The results, displayed by Uniform Manifold Approximation and Projection (UMAP) dimensional reduction, showed that the ZsGreen-tracked progenitor cell population includes phenotypic subsets consistent with HSPCs (**Figure 1I**). Each ZsGreen$^+$ progenitor population represented a similarly small proportion of the corresponding non-tracked progenitor cell population (**Figure 1J**). These results support the existence of EHT in the adult mouse contributing to generation of hematopoietic progenitors and mature cells.

## Adult BM ECs cultured ex vivo generate transplantable HSPCs

To investigate whether adult BM ECs can generate hematopoietic cells ex vivo, we cultured BM cells isolated from tamoxifen-treated Cdh5-Cre/ZsGreen mice. Initially, BM single-cell (sc) suspensions were cultured under two conditions (**Figure 2A**): (1) on 'Primaria' pro-adhesive flasks, and (2) on OP9 stromal cell (**Nakano, 1996**) monolayers grown on gelatin-coated conventional tissue culture flasks. To attempt recreating BM niches, fresh wild-type (WT) BM cells were added to the ZsGreen-tracked BM cell cultures twice/week throughout the culture period (see Methods

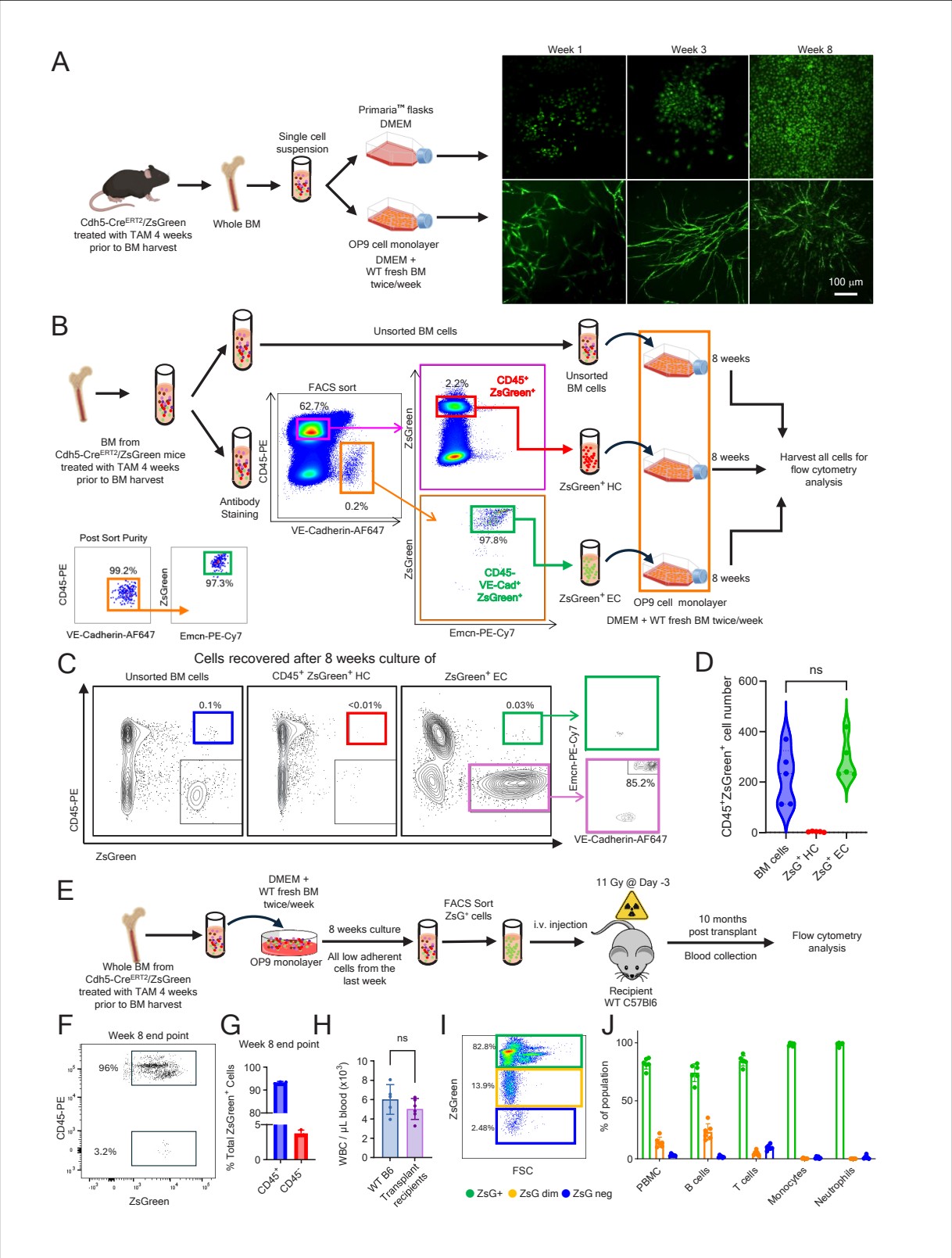

**Figure 2.** Bone marrow (BM) endothelial cells (ECs) generate engraftable hematopoietic cells ex vivo. (**A**) BM cells from tamoxifen-treated mice were cultured on high-attachment Primaria flasks or OP9 cell monolayers. Representative images show ZsGreen+ cells at weeks 1, 3, and 8. (**B**) Workflow for culturing unsorted and sorted BM cell populations. Post-sort purity of ZsGreen+ ECs is shown in the bottom left panel. All cells s were cultured (8 weeks) on OP9 cell monolayers supplemented with WT BM cells. Culture medium and floating cells were removed twice/week for 7 weeks. At the start of week

*Figure 2 continued on next page*

*Figure 2 continued*

8, one final WT BM and medium supplementation was implemented prior to harvest at the end of week 8. Representative flow cytometry plots (**C**) and quantification (**D**) of CD45+ZsGreen+ cells from each of the 8 week cultures (*n* = 5). (**E**) Floating/loosely adherent ZsGreen+ cells from unsorted BM 8-week cell cultures were sorted and transplanted ($5 \times 10^4$, $2.5 \times 10^4$, $1.25 \times 10^4$, or $6.25 \times 10^3$ cells) into lethally irradiated (11 Gy) WT mice (*n* = 2/group). Representative flow cytometry image (**F**) and quantification (**G**) of low-adherent cells harvested after 8 weeks of culture, showing that >95% (group average) of ZsGreen+ low-adherent cells are CD45+. These ZsGreen+CD45+ cells were sorted for transplantation. White blood cell (WBC) counts from five control mice (no irradiation or transplant) (**H**) and percent ZsGreen+, ZsGreen dim, and ZsGreen− cells (**I, J**) in blood of transplant recipients 10 months post-transplant (*n* = 6). Dots represent individual mice. Data are shown as mean ± SD. ns, not significant by Student's *t*-test.

The online version of this article includes the following figure supplement(s) for figure 2:

**Figure supplement 1.** Bone marrow (BM) endothelial cells (ECs) generate engraftable hematopoietic cells ex vivo.

for details). By week 8, BM cells cultured on 'Primaria' dishes formed a confluent monolayer of ZsGreen-tracked cells with a typical EC morphology, whereas BM cells cultured on OP9 monolayers exhibited a fibroblast-like morphology with the ZsGreen+ cells clustering in foci, without forming a monolayer (***Figure 2A***). Sorted ZsGreen+ ECs cultured for 4 weeks on OP9 monolayer also exhibited a fibroblast-like morphology (***Figure 2—figure supplement 1A***). We visualized some round ZsGreen-tracked (ZsGreen+) cells in cultures of BM cells grown onto OP9 monolayers supplemented with fresh BM cells, but not in cultures of BM cells grown onto 'Primaria' surfaces supplemented with fresh BM cells (***Figure 2—figure supplement 1B***). This hinted that the OP9 and BM cell culture system may allow hematopoietic cell emergence from Cdh5-Cre/ZsGreen-tracked EC ex vivo.

To test this possibility, we generated a pool of BM cells (from 20 adult mice; age 10–18 weeks) and used the pool to derive 3 populations: sorted BM CD45−VE-Cadherin+ZsGreen+ ECs; sorted CD45+ZsGreen+ hematopoietic cells; and unsorted BM cell populations (***Figure 2B***). We then cultured these three cell populations under the same conditions (OP9 monolayer and supplementation with fresh BM cells twice/week). After 8-week culture, flow cytometry analysis detected CD45+ZsGreen+ cells from the unsorted BM and the sorted CD45−VE-Cadherin+ZsGreen+ cell cultures. However, CD45+ZsGreen+ were virtually absent from the CD45+ZsGreen+ cell cultures (***Figure 2C, D***), likely attributable to the culture conditions not designed to support growth, differentiation or survival of hematopoietic cells. These results show that CD45−VE-Cadherin+ZsGreen+ ECs can generate CD45+ZsGreen+ ex vivo. Notably, most (>80%) CD45−ZsGreen+ cells retained expression of VE-Cadherin and Endomucin, thereby confirming their endothelial identity (***Figure 2C***, most right panel). Importantly, the virtual absence of CD45+ZsGreen+ cells in 8-week cultures of sorted CD45+ZsGreen+ cells shows that pre-existing CD45+ZsGreen+ hematopoietic cells, derived from tamoxifen-dependent and -independent processes, are effectively removed during the extended culture. This further suggests that CD45+Zs-Green+ hematopoietic cells, potentially contaminating the sorted ECs, are unlikely contributors to EC-derived hematopoiesis. Rather, these results show that the ECs are the likely cell source of hematopoietic cells in this ex vivo co-culture model.

Next, we tested whether the ex vivo-derived CD45+ZsGreen+ cells are functional in lethally irradiated recipients (***Figure 2E***). To this end, we obtained unfractionated BM cells from 50 tamoxifen-treated Cdh5-Cre/ZsGreen mice, cultured the cells onto OP9 monolayers with fresh BM supplementation, sorted the ZsGreen+ cells (>95% of which were CD45+ by flow cytometry) at the end of 8-week culture, and transplanted these cells into eight lethally irradiated WT recipients; $5 \times 10^3$ cells; $2.5 \times 10^3$ cells; $1.25 \times 10^3$ cells; $6.25 \times 10^2$ cells, *n* = 2/group (***Figure 2E–G***). Two mice (recipients of $6.25 \times 10^2$ and $1.25 \times 10^3$ cells) died on days 4 and 6, but the remaining six mice survived and remain well at the time of manuscript revision (10 months post-transplant). Ten months post-transplant, peripheral blood white blood cell (WBC) counts were within normal range in all surviving transplant recipients, indicative of hematopoietic reconstitution (***Figure 2H***). Flow cytometry revealed that 81.9% of PBMC were ZsGreen+ (***Figure 2I, J***). Most neutrophils (98.6%), monocytes (98.5%), B cells (74.1%), and T cells (84.3%) were ZsGreen+.

These results confirm that BM ECs propagated ex vivo onto OP9 monolayers with BM cell supplementation produce hematopoietic cells and show that this output includes HSPC capable of engrafting and generating hematopoietic cell progeny.

## Adult BM ECs can give rise to hematopoietic cells following transfer into conditioned recipients

Next, we examined whether adult BM ECs are hemogenic in transplant recipients. To this end, we FACS-sorted CD45⁻VE-Cadherin⁺ZsGreen⁺ ECs from the BM of tamoxifen-pretreated (4 weeks prior to BM harvest) Cdh5-Cre^ERT2(PAC)/ZsGreen mice (*Figure 3A*; *Figure 3—figure supplement 1A, B*) and transplanted these cells into adult WT C57Bl/6 unconditioned mice (PBS) or conditioned by fluorouracil (5-FU treated) prior to transplant (*Xu et al., 2018*). The choice of 5-FU conditioning rather than lethal irradiation of adult mice was driven by previous experiments showing the difficulties at reconstituting lethally myelo-ablated adult recipients with hemogenic yolk sac cells, which reconstituted conditioned newborns (*Yoder et al., 1997*). Four weeks after transplant, we observed a significant increase in the proportion of ZsGreen⁺ ECs in the BM and ZsGreen⁺CD45⁺ hematopoietic cells in the BM and peripheral blood of the transplant recipients conditioned by 5-FU but not controls (PBS) (*Figure 3B, C*; *Figure 3—figure supplement 1C*). These CD45⁺ZsGreen-tracked cells in BM and blood included granulocytes, monocytes, and lymphocytes (*Figure 3D*). Thus, BM-derived CD45⁻VE-Cadherin⁺ZsGreen⁺ cells transferred into 5-FU-conditioned recipients gave rise to detectable CD45⁺ZsGreen⁺ hematopoietic cells.

We further examined the effect of mouse age on the endothelial hemogenic potential by treating the mice with tamoxifen between weeks 6 and 32 of age. Four weeks later (weeks 10–36 of age), we measured the percentage of ZsGreen-tracked CD45⁺ hematopoietic cells in the BM. We observed that tamoxifen inductions beyond week 10 of age resulted in a progressive decrease of the CD45⁺ hematopoietic cell output and detected an inverse correlation (Pearson's $r$ –0.63, $p < 0.0001$) between age and EC hemogenic potential (*Figure 3E*). This progressive decline of CD45⁺ cell output was not coupled with a loss of BM EC fluorescence, since virtually all BM ECs were ZsGreen⁺ throughout the duration of the experiment (*Figure 3F*), consistent with the stability of Cre-mediated labeling of Cdh5-expressing cells (*Wang et al., 2013*). These observations suggest an age-related loss of hemogenic capability of BM ECs.

Additionally, we examined whether tracked hematopoietic cells from adult BM EC are functional. Since lymphocyte trafficking from the peripheral blood to the peritoneal cavity is critical for their function at this site (*Auffray et al., 2009*; *Ghosn et al., 2010*), we first evaluated the spontaneous migration of tracked CD45⁺ hematopoietic cells to the peritoneal cavity. Compared to no-tamoxifen controls, tamoxifen-treated Cdh5-Cre^ERT2(PAC)/ZsGreen mice displayed a significant increase of CD45⁺ZsGreen⁺ monocytes, macrophages, and B1, B2, and T lymphocytes in the peritoneal cavity (*Figure 3—figure supplement 1D, E*).

After inducing peritonitis with thioglycolate (TGL, 4 hr) in tamoxifen-treated Cdh5-Cre^ERT2(PAC)/ZsGreen mice, the overall number of peritoneal leukocytes increased substantially compared to untreated (PBS) controls (*Figure 3G*), mostly attributable to neutrophils (*Figure 3H*). In addition, the peritoneal ZsGreen⁺ and ZsGreen⁻ cell populations exhibited a similar cell type distribution in TGL-treated mice (*Figure 3I*). We further examined *Escherichia coli* (K-12 strain) phagocytosis and reactive oxygen species (ROS) production in peritoneal cell exudates in response to TGL (*Figure 3J–O*). Both ZsGreen⁺ and ZsGreen⁻ peritoneal neutrophils and macrophages comparably phagocytosed *E. coli* and generated ROS (*Figure 3J, K, M, N*), except that the phagocytic and ROS production of ZsGreen⁺ macrophages was somewhat higher than that of ZsGreen⁻ macrophages (*Figure 3L, O*).

Thus, Cdh5-tracked mature neutrophils and macrophages are functional at trafficking and homing and presumably capable of contributing to the host response to tissue inflammation.

## Adult EHT is independent of preexisting hematopoietic cell progenitors

Faithful Cre-reporter lineage tracing requires that Cre recombinase activity be restricted to the intended cell type (*Álvarez-Aznar et al., 2020*). In our system, Cdh5-Cre^ERT2 is expected to drive recombination specifically in ECs, as *Cdh5* is an established EC-specific marker. However, flow cytometry has occasionally revealed rare VE-Cadherin⁺CD45⁺ in mouse BM, exemplified in *Figure 3—figure supplement 1C*, potentially reflecting double-positive cells. To address the possibility that cells co-expressing VE-Cadherin/*Cdh5* and CD45/*Ptprc* exist in the mouse BM, we analyzed publicly available sc-RNAseq data from adult mouse BM (*Kucinski et al., 2024*). This analysis showed that only a small subset of plasmacytoid dendritic cells (pDCs) co-express VE-Cadherin and CD45, but not HSPC or other mature blood cells (*Figure 4—figure supplement 1A–D*). pDCs are terminally differentiated

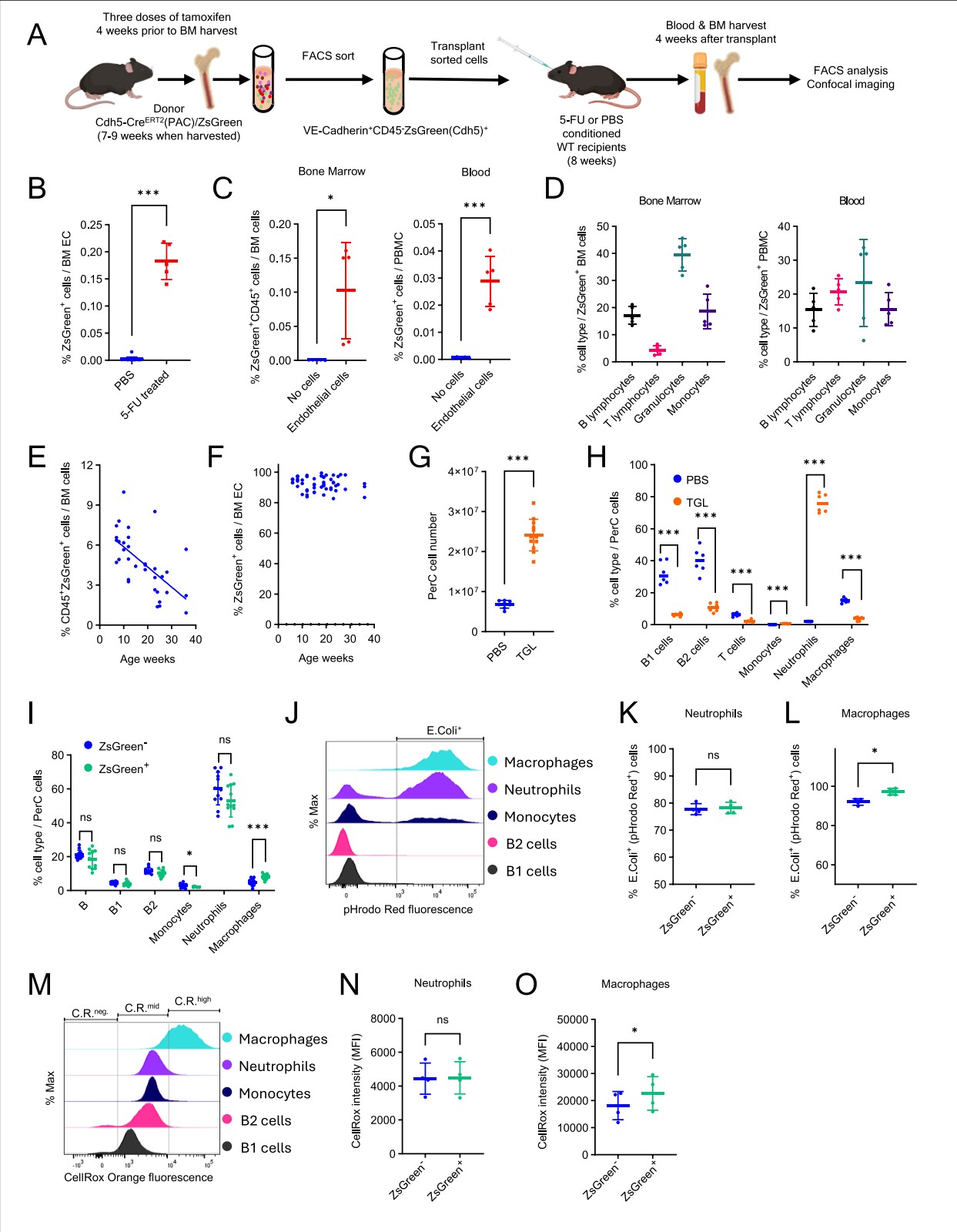

**Figure 3.** Adult bone marrow (BM) endothelial cells (ECs) give rise to hematopoietic cells following transfer into conditioned recipients. (**A**) Transplant experiment: donor ECs from BM of tamoxifen-treated mice were FACS-sorted and transplanted into WT C57Bl/6 recipients conditioned with 5-FU or PBS. (**B**) ZsGreen$^+$ ECs detected in BM of 5-FU-conditioned ($n$ = 5) or PBS-conditioned ($n$ = 15) recipients of ECs 4 weeks post-transplant. ZsGreen$^+$CD45$^+$ cells (**C**) and cell type distribution (**D**) in the BM and blood of 5-FU-conditioned transplant recipients of BM ECs or no cell controls

*Figure 3 continued on next page*

Figure 3 continued

($n$ = 5/group). Age-dependent decline of ZsGreen$^+$CD45$^+$ cells (E) but not ZsGreen$^+$VE-Cadherin$^+$ cells (F) in the BM of Cdh5-Cre$^{ERT2}$(BAC)/ZsGreen mice ($n$ = 35) treated with tamoxifen 4 weeks prior to harvest. Cell number (G; mouse $n$ = 8–12) and cell type distribution (H; mouse $n$ = 6) in the peritoneal cavity (PerC) of PBS- or thioglycolate (TGL)-pretreated (4 hr) mice. (I) ZsGreen$^+$ and ZsGreen$^-$ PerC cell types in TGL-pretreated mice ($n$ = 12). (J) Representative histograms depicting pHrodo Red fluorescence detection of *E. coli* phagocytosis. *E. coli*$^+$ phagocytosis by ZsGreen$^+$ and ZsGreen$^-$ PerC neutrophils (K) and macrophages (L) in TGL-pretreated mice ($n$ = 4). (M) Representative histograms depicting CellRox Orange fluorescence for cell-associated ROS detection. CellRox mean fluorescence intensity (MFI) in ZsGreen$^+$ and ZsGreen$^-$ PerC neutrophils (N) and macrophages (O) in TGL-pretreated mice ($n$ = 4). Dots represent individual mice. Data are shown as mean ± SD. *$p < 0.05$, ***$p < 0.001$, ns, not significant by Student's $t$-test.

The online version of this article includes the following figure supplement(s) for figure 3:

**Figure supplement 1.** Adult bone marrow (BM) endothelial cells (ECs) give rise to hematopoietic cells following transfer into conditioned recipients.

cells and unlikely progenitors of tracked CD45$^+$ multilineage progeny in our Cdh5-cre mice. Nonetheless, it is plausible that other, currently unidentified, hematopoietic cells may also co-express *Cdh5*/VE-Cadherin and *Ptprc*/CD45 and possess functional Cre$^{ERT2}$ activity, inducing fluorescence in these cells upon tamoxifen administration.

To address these possibilities, we transplanted lethally irradiated (11 Gy) WT C57Bl6 mice ($n$ = 6) with ZsGreen$^-$Lin$^-$Sca1$^+$cKit$^+$ (LSK) progenitors (>99% purity; *Figure 4—figure supplement 1E–G*) from tamoxifen untreated Cdh5-Cre$^{ERT2}$(PAC)/ZsGreen mice and examined whether these ZsGreen$^-$ hematopoietic progenitors can become fluorescent after tamoxifen administration (*Figure 4A*). As a positive control, we also transplanted lethally irradiated WT C57Bl6 mice ($n$ = 2) with an LSK population enriched for ZsGreen$^+$ cells 45.9% purity, with the remaining 54.1% comprising ZsGreen$^-$ LSK cells (*Figure 4A*, *Figure 4—figure supplement 1F*). Four weeks post-transplant, we administered tamoxifen and monitored the peripheral blood for the presence of ZsGreen$^+$ hematopoietic cells over 6 months.

Expectedly, all LSK recipients (ZsGreen$^-$ LSKs or LSK enriched with ZsGreen$^+$ cells) showed successful hematopoietic reconstitution as evidenced by normal blood WBC counts at 10 weeks post-transplant (*Figure 4B*). Importantly, the six mice transplanted with ZsGreen$^-$ LSKs did not produce ZsGreen$^+$ PBMCs post-tamoxifen administration, indicating that the transplanted LSKs and their progeny did not express tamoxifen-inducible Cdh5-Cre$^{ERT2}$ recombinase activity (*Figure 4C, D*). Instead, the two mice transplanted with ZsGreen$^+$-enriched LSKs displayed a similar percent of ZsGreen$^+$ PBMCs prior to and after tamoxifen administration (*Figure 4D*). Collectively, these results demonstrate that LSK progenitors in adult BM lack tamoxifen-inducible Cdh5 expression and do not contribute to tamoxifen-induced adult EHT in our Cdh5-reporter mice. Rather, these results strongly support the conclusion that Cdh5$^+$CD45$^-$ ECs are a source of hematopoietic cells in the adult mouse.

In additional experiments, we took advantage of the relative insensitivity of BM EC subsets to irradiation relative to BM hematopoietic cells (*Skulimowska et al., 2024*) to examine the possibility that ECs surviving after lethal irradiation may be hemogenic. As a lethal dose of irradiation effectively eliminates HSPCs and requires hematopoietic reconstitution for survival, we transplanted WT (untracked) BM cells into lethally irradiated Cdh5-Cre/mTmG mice and treated the mice with tamoxifen 4 weeks after transplantation (*Figure 4E*). Prior to tamoxifen administration, >99% of PBMCs were not fluorescent, indicating that these cells derived from the transplanted WT BM rather than host-derived (*Figure 4F*). After tamoxifen treatment, a progressive increase in EGFP$^+$ PBMCs was observed, reaching ~0.55% by 6 weeks, which included myeloid cells and B and T lymphocytes (*Figure 4F, G*). Although we cannot exclude the possibility that rare EGFP$^+$ hematopoietic progenitor (tracked tamoxifen dependently or independently) may have survived the irradiation, the presence of EGFP$^+$ hematopoietic cells in the circulation of lethally irradiated Cdh5-Cre/mTmG mice suggests their derivation from radioresistant Cdh5$^+$ ECs rather than from radiosensitive hematopoietic progenitors. These results further support the view that adult ECs possess hemogenic potential and can produce hematopoietic cells in vivo.

## Single-cell tracking confirms the presence of hemogenic ECs in adult BM

To directly trace hematopoietic cell progeny arising from individual adult ECs, we exploited the PolyloxExpress sc genetic barcoding system. We generated Cdh5-Cre$^{ERT2}$/ZsGreen/PolyloxExpress mice, in which both the ZsGreen and Polylox transgenes are inserted in the Rosa26 locus, such that

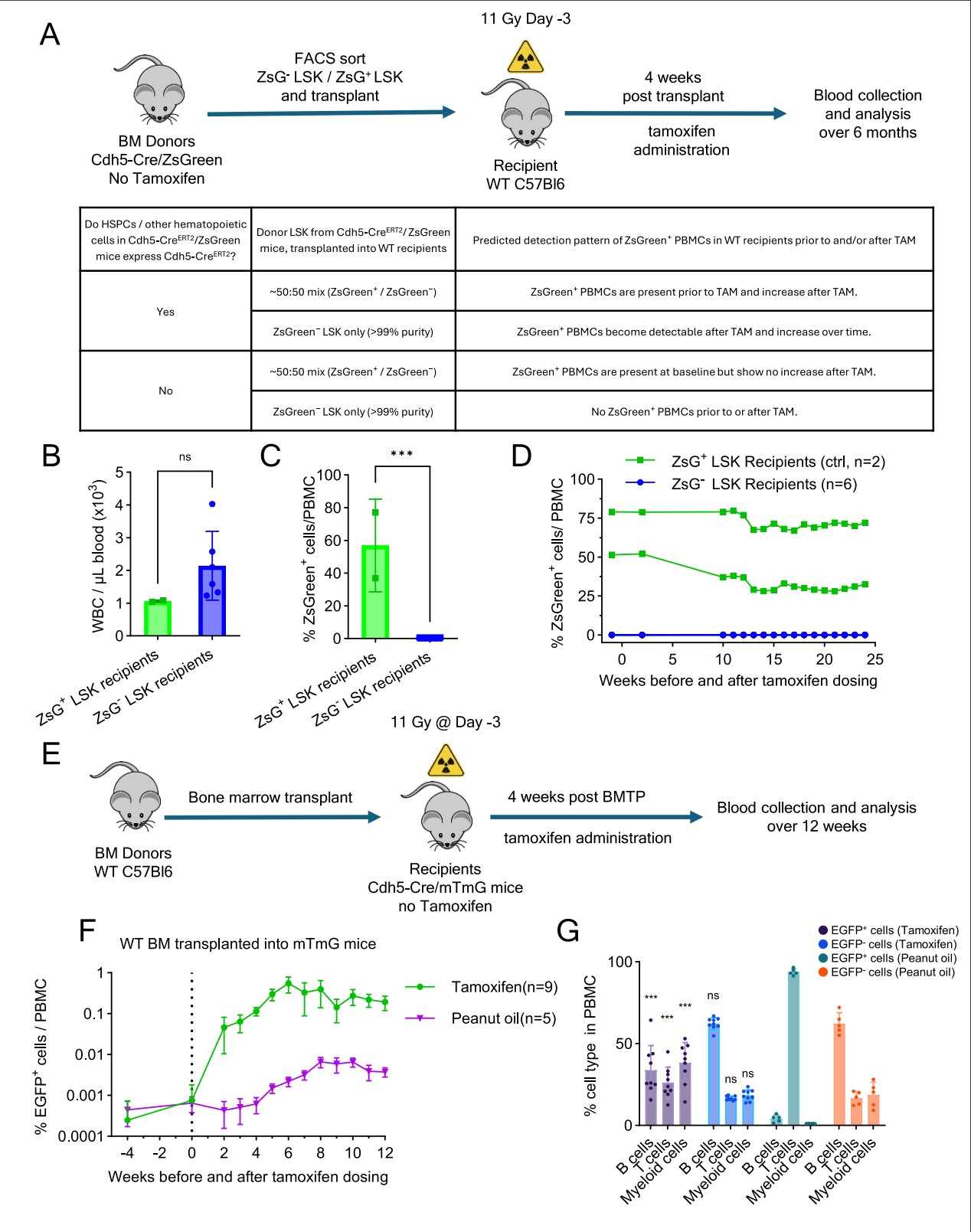

**Figure 4.** Independence of adult endothelial-to-hematopoietic transition (EHT) from preexisting hematopoietic stem and progenitor cell (HSPC). (**A**) Transplantation experiment: donor LSK sorting, recipient irradiation, transplantation, tamoxifen treatment, and analysis (top). Tabular representation of possible outcomes of the experiment designed to address the question 'Do HSPCs/other hematopoietic cells in Cdh5Cre$^{ERT2}$/ZsGreen mice express Cdh5-Cre$^{ERT2}$?' (bottom). Blood WBC counts (**B**), percent ZsGreen$^+$ peripheral blood mononuclear cell (PBMC) (**C**), and time course of ZsGreen$^+$ PBMC

*Figure 4 continued on next page*

*Figure 4 continued*

detection (**D**) in transplant recipients of ZsGreen⁻ LSK (5 × 10⁴ or 2.5 × 10⁴ cells/mouse; *n* = 3/group) and ZsGreen-enriched LSKs (2.8 × 10³ cells/mouse; *n* = 2). Results in B and C are from week 24 post-tamoxifen. (**E**) Experiment: WT BM transplantation (BMTP) into lethally irradiated Cdh5-Cre/mTmG mice (*n* = 9). Four weeks later, tamoxifen was administered; blood was monitored for 16 weeks. EGFP⁺ PBMC detection before and after tamoxifen or peanut oil administration (**F**) and cell type distribution of EGFP⁺ and EGFP⁻ PBMCs at week 12 post-tamoxifen or peanut oil (**G**) in Cdh5-Cre/mTmG recipients (*n* = 9) of WT BM (5 × 10⁶ cells). Statistical significance reflects comparisons between EGFP⁺ and EGFP⁻ cells in the tamoxifen versus peanut oil groups. Dots represent individual mice. Data are shown as mean ± SD. ***p < 0.001, ns, not significant by Student's *t*-test.

The online version of this article includes the following figure supplement(s) for figure 4:

**Figure supplement 1.** Bone marrow (BM) plasmacytoid dendritic cells (pDCs), but not other BM hematopoietic cells, express *Cdh5* and *Ptprc*, encoding CD45.

individual ZsGreen⁺ cells contain a single Polylox barcode (***Pei et al., 2020***; ***Pei et al., 2017***; ***Pei et al., 2019***). To evaluate EC-derived hematopoietic cell output, we harvested BM from tamoxifen-treated Cdh5-Creᴱᴿᵀ²/ZsGreen/PolyloxExpress mice (*n* = 3), sorted the ZsGreen⁺VE-Cadherin⁺Endomucin⁺ (purity >95%) and the ZsGreen⁺VE-Cadherin⁻Endomucin⁻CD45⁺ hematopoietic cells (purity >98%), mixed these populations (1:1 ratio; total 147,446 cells) and processed for 10x Illumina Sequencing and PacBio sequencing (detailed in Methods). We recovered the sc transcriptome from 93,553 cells; of these, 4069 cells had a barcode (***Figure 5A***, details in Methods).

Unsupervised clustering of sc transcriptome data revealed 34 clusters, 31 of which remained after doublet removal (***Figure 5B***; ***Figure 5—figure supplement 1A–D***). Cell clusters 0, 1, 22, and 13 were annotated as 'Endothelial cells' based on expression of the classical EC markers *Cdh5*, *Pecam1*, *Kdr*, and *Flt1*, and absence of *Ptprc*/CD45, *Runx1*, and the mesenchymal cell markers *Cxcl12*, *Lepr*, *Pdgfrb*, and *Col1a2* expression (***Figure 5B***; ***Figure 5—figure supplement 1E***). Cell cluster 14 was annotated as 'Mesenchymal type' based on co-expression of *Cxcl12*, *Lepr*, *Pdgfrb*, *Col1a2*, but expressed *Runx1* and the EC markers *Cdh5* and *Pecam1* (***Figure 5B***; ***Figure 5—figure supplement 1E***). The remaining cell clusters included the hematopoietic progenitors and mature blood cells of various lineages (***Figure 5B***; ***Figure 5—figure supplement 1E***). Creᴱᴿᵀ² transcripts were detected exclusively in Cdh5-expressing EC populations and were absent from Ptprc/CD45-expressing hematopoietic cells (***Figure 5—figure supplement 1E***). Creᴱᴿᵀ² expression levels in ECs were low but closely tracked with the expression patterns of canonical endothelial markers (*Cdh5*, *Pecam1*, *Emcn*, and *Eng*).

Barcode analysis revealed robust *Polylox* barcode diversity among cell populations, including 274 'true' barcodes, defined as barcodes with low generation probability (***Figure 5—figure supplement 2A***, p $_{gen}$ < 1 × 10⁻⁶) (***Pei et al., 2017***) consistent with rare, unique recombination events (***Figure 5—figure supplement 2A–C***). Expectedly, 'true' barcodes linked HSPC to downstream progenitors and mature hematopoietic cells, validating the system (***Figure 5C***).

Notably, 169 of 828 ECs (from 'endothelial' cell clusters 0, 1, 22, and 13) were marked with 'true' barcodes. The results detected 19 links between EC and hematopoietic cells based on their shared 'true' barcode (***Figure 5D***). These hematopoietic cells linked to ECs by shared 'true' barcodes included HSPC, EPC, GMP, and mature blood cells, encompassing granulocytes, monocytes, dendritic cells, B and T lymphocytes, plasma cells, and pDCs (***Figure 5D, E***). These results provide direct evidence, at sc resolution, that adult mouse BM ECs can generate hematopoietic progenitors and mature blood cells.

Additionally, 26 'Mesenchymal type' cells (from cluster 14; co-expressing mesenchymal cell markers, *Runx1*, and the EC markers *Cdh5* and *Pecam1*) were also marked by 11 'true' barcodes, 8 of which were shared with hematopoietic cell progenitors (HSPC and Erythroid) and mature blood cells (***Figure 5E, F***). Also, four 'true' barcodes linked 'Mesenchymal type' cells to ECs (from clusters 0, 1, 22, and 13), suggesting either a shared precursor or derivation from each other. Although it cannot be excluded that barcoding missed identification of mesenchymal cell links to other ECs or cells, these results raise the possibility that certain BM 'Mesenchymal type' cells may produce hematopoietic cell progeny. Despite similarities in sc tracing results linking ECs and 'Mesenchymal type' cells to hematopoietic cells, these two cell populations display a distinctive transcriptome profile (***Figure 5G***; ***Figure 5—figure supplement 1E***).

Additional analysis of all tracked cells showed that several ECs and, to a lower extent, 'Mesenchymal type cells' shared 'true' barcodes (***Figure 5—figure supplement 2C***), indicating clonal expansion. Transcriptome-based sc cell cycle analysis confirmed the presence of ECs and 'Mesenchymal

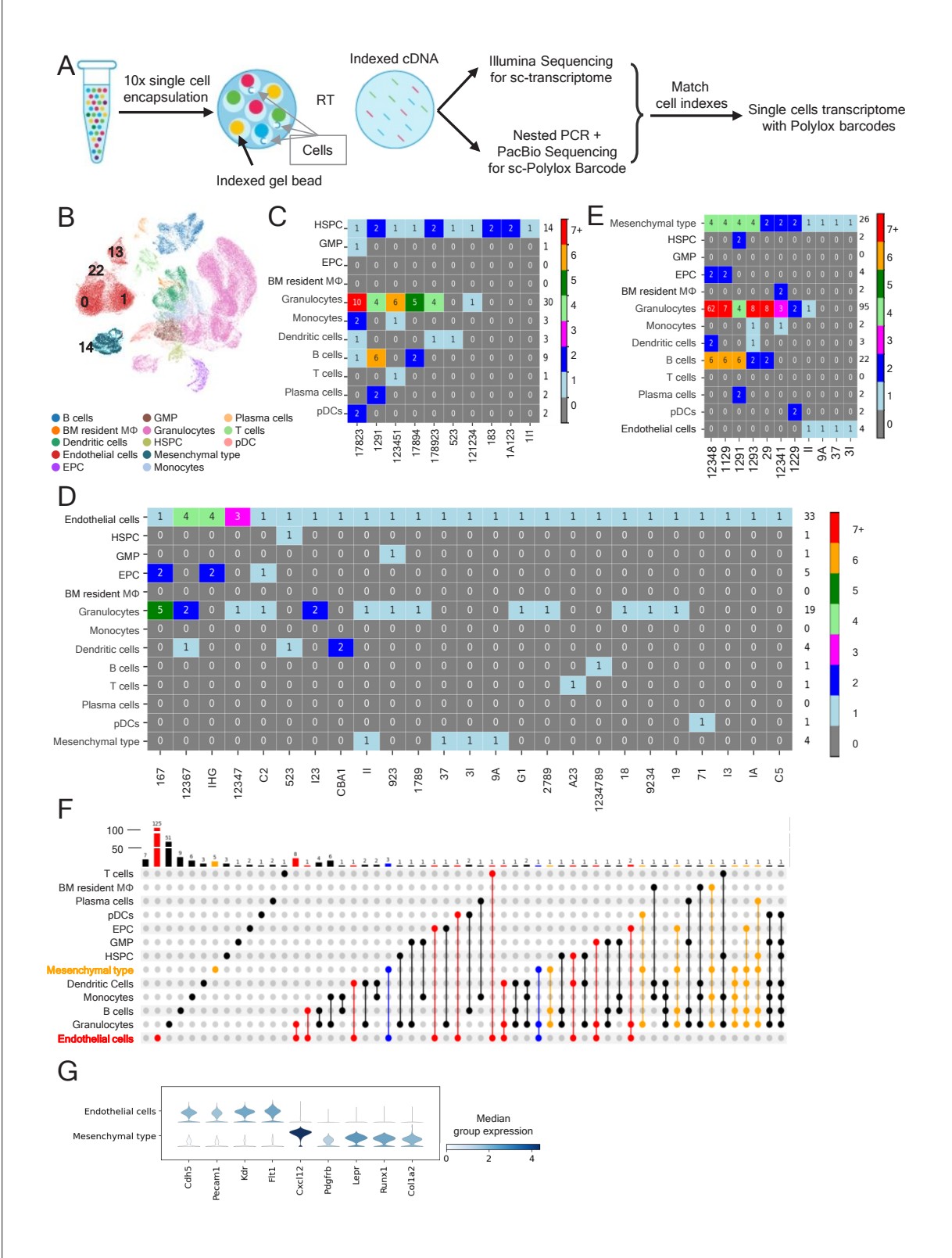

**Figure 5.** Polylox sc lineage tracing links adult bone marrow (BM) endothelial cells (ECs) to hematopoietic progenitors and mature blood cell progeny. (**A**) Schematic of Polylox barcode and transcriptome profiling. FACS-enriched ECs (ZsGreen⁺VE-Cadherin⁺Endomucin⁺) and EC-depleted (ZsGreen⁻VE-Cadherin⁻Endomucin⁻) BM cells from tamoxifen-treated Cdh5-Creᴱᴿᵀ²/ZsGreen/PolyloxExpress mice (n = 3, 10-week-old at the time of tamoxifen treatment) were mixed (1:1), and encapsulated (147,446 cells loaded; 93,553 processed). Indexed cDNA was used for scRNA-seq and barcode detection

*Figure 5 continued on next page*

*Figure 5 continued*

by PacBio sequencing after nested PCR enrichment; barcode-transcriptome integration was accomplished via shared cell indices. (**B**) Uniform Manifold Approximation and Projection (UMAP) clustering and cell type annotation. Clusters 0, 1, 13, and 22 comprise ECs; cluster 14 comprises Mesenchymal-type cells. Heatmaps showing 'true' Polylox barcodes (pGen <1 × 10$^{-6}$) linking hematopoietic stem and progenitor cells (HSPCs) to hematopoietic cells (**C**), ECs to hematopoietic and other cells (**D**), and Mesenchymal-type cells to other cells (**E**). The numbers within the colored boxes identify cell number; the labels at the bottom of each column denote the barcode shared by all cells in that column; the number on the right side of the heatmaps reflects the total number of cells in each row. (**F**) UpSet plot showing cells (identified by colored dots) sharing the same 'true' barcode (identified by lines connecting the colored dots); bar graph at the top of the plot reflects (height and number on each bar) the number of 'true' barcodes. Colors of dots: EC (red), Mesenchymal-type (orange), ECs connecting with Mesenchymal-type cells (blue), cells other than ECs and Mesenchymal-type cells (black). (**G**) Violin plots showing selected gene expression profile in Mesenchymal-type cells (cluster 14) and ECs (clusters 0, 1, 13, 22 combined).

The online version of this article includes the following figure supplement(s) for figure 5:

**Figure supplement 1.** Single-cell RNA-seq analysis of bone marrow (BM) ZsGreen$^+$ cells from tamoxifen-treated Cdh5-Cre/ZsGreen/Polylox mice.

**Figure supplement 2.** Identification and distribution of 'True' Polylox barcodes across cell types.

type cells' in the S and G2/M phases, albeit to a much lower degree than HSPC (*Figure 5—figure supplement 2D*).

Together, these results demonstrate at a sc level that adult BM ECs can generate hematopoietic cell progeny of HSPC and mature blood cells. The results further raise the possibility that 'Mesenchymal type' cells, marked by a hybrid endothelial and stromal phenotype, may represent an additional source of hemogenic activity in adult BM.

## Single-cell transcriptome identifies a *Cdh5*$^+$*Col1a2*$^+$ *Runx1*$^+$ cell population in the adult BM

To further characterize adult hemogenic cell populations, we analyzed publicly available scRNAseq datasets comprising BM cells from adult mice (1–16 months of age) (*Baryawno et al., 2019*; *Tikhonova et al., 2019*; *Baccin et al., 2020*; *Sivaraj et al., 2021*; *Zhong et al., 2020*; *Lang et al., 2025*; *Smith et al., 2025*) and embryonic caudal artery ECs (9.5–11.5 days post coitum) (*Zhu et al., 2020*). After quality control and dimensional reduction, the remaining 434,810 cells clustered into 71 distinct populations (*Figure 6A–D*). Among the *Cdh5*-expressing endothelial clusters, two clusters, cluster 8 composed predominantly of embryonic cells (98%) and cluster 50 composed largely of adult BM-derived cells (95.5%) (*Figure 6E*), were notable in comprising cells co-expressing *Cdh5* and *Runx1*, a transcription factor that marks the hemogenic EC identity during development (*Howell et al., 2021*; *Figure 6B*).

We jointly analyzed the transcriptome from the publicly available embryonic cluster 8 and adult cluster 50 (*Figure 6A–C*), and from our sc Polylox dataset, including adult clusters 14 (Mesenchymal type) and adult EC clusters 0, 1, 13, and 22 (*Figure 5B*). All populations expressed canonical EC markers (*Cdh5*, *Pecam1*, *Kdr*, and *Flt1*), though at different levels, but expression of *Runx1* was mostly confined to clusters 8, 50, and Polylox 14 (*Figure 6F*). Interestingly, adult clusters 50 and Polylox 14 co-expressed the mesenchymal cell-associated genes *Col1a2*, *Lepr*, *Cxcl12*, and *Pdgfrb*, distinguishing these two clusters from the embryonic cluster 8 and adult Polylox clusters 0, 1, 13, and 22 (*Figure 6F*). Additionally, embryonic cluster 8 exhibited higher expression of EHT-related transcription factors (FLI1, LMO2, TAL1, and ERG) (*Serina Secanechia et al., 2022*; *Bergiers et al., 2018*; *Guibentif et al., 2017*; *Wilson et al., 2010*), some of which were variably expressed by the Polylox clusters 0, 1, 13, and 22 (*Figure 6F*).

To explore whether cells expressing *Cdh5*, *Col1a2*, and *Runx1* are unique to BM (*Figure 6G*), we looked at other adult mouse tissues. Analysis of public scRNAseq datasets from 11 adult mouse tissues (*Kalucka et al., 2020*) showed that *Cdh5*, *Runx1*, and *Col1a2* expressing ECs are largely restricted to the BM, with only 3 such cells detected out of ~32,000 ECs from other tissues (*Figure 6H, I*; *Kalucka et al., 2020*). These observations, together with the Polylox sc tracing experiments, indicate that adult mouse BM harbors a cell population with mixed endothelial and mesenchymal phenotype marked by *Runx1*, *Col1a2,* and *Cdh5* expression and raises the possibility that this population has hemogenic potential.

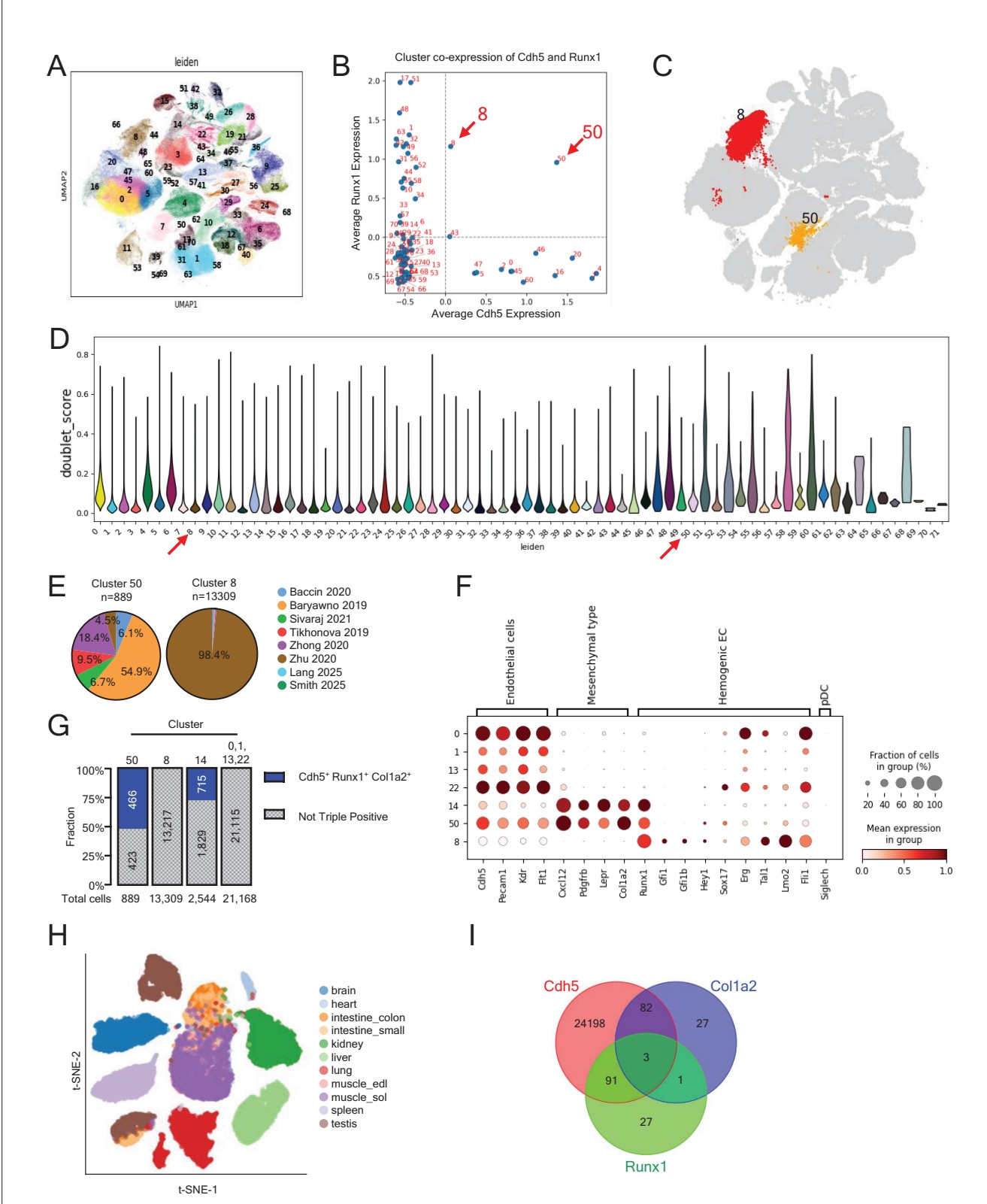

**Figure 6.** Sc transcriptomic analysis of prospective hemogenic endothelial cells (ECs). (**A**) Uniform Manifold Approximation and Projection (UMAP) clustering of 434,810 cells from eight public scRNA-seq datasets. (**B**) Dot plot showing relative Cdh5 and Runx1 co-expression across clusters; clusters 8 and 50 co-express both genes. (**C**) UMAP highlighting clusters 8 and 50; all other clusters shown in grey. (**D**) Violin plots of doublet scores across Leiden clusters. Clusters 50 and 8 show no evidence of doublet enrichment. (**E**) Datasets proportional contribution to clusters 50 and 8; each dataset is

*Figure 6 continued on next page*

*Figure 6 continued*

color-coded. (F) Dot plot showing expression of selected marker genes in clusters 50 and 8 (from the public sc RNA-seq datasets listed in *Figure 7D*) and from clusters 0, 1, 13, 22, and 14 (from Polylox scRNA-seq; *Figure 5B*). Results reflect mean expression and fraction of cells in group. (G) Cdh5, Runx1, and Col1a2 co-expression in the indicated clusters as a fraction of cells in the cluster. (H, I) t-SNE plot of ECs from 11 murine tissues (G) and Venn diagram (H) showing rare co-expression of Cdh5, Runx1, and Col1a2 in these tissues.

## *Col1a2* and *Runx1* expression in BM ECs

To evaluate a potential contribution of *Col1a2* expression to EHT in adult BM, we generated the mouse lines Col1a2-Cre^ERT2/mTmG and Col1a2-Cre^ERT2/ZsGreen to track cells expressing the *Col1a2* gene (*Figure 7—figure supplement 1A*). Four weeks after tamoxifen administration, the BM of Col1a2-Cre^ERT2/ZsGreen mice contained abundant ZsGreen$^+$ cells (*Figure 7—figure supplement 1B*). By flow cytometry, a subset of these BM Col1a2-tracked ZsGreen$^+$ cells was VE-Cadherin$^+$CD45$^-$ and RUNX1$^+$ consistent with an endothelial identity and perhaps hemogenic potential (*Figure 7—figure supplement 1C*). Noteworthy, a similar subset of VE-Cadherin$^+$CD45$^-$RUNX1$^+$ cells was detected in BM of adult WT C57Bl/6 mice (*Figure 7—figure supplement 1D*) and in BM of Cdh5-Cre^ERT2(PAC)/ZsGreen mice after tamoxifen treatment (*Figure 7—figure supplement 1E*).

To evaluate hemogenic potential, we looked for Col1a2-tracked CD45$^+$ hematopoietic cells in Col1a2-Cre^ERT2/mTmG and Col1a2-Cre^ERT2/ZsGreen mouse lines after tamoxifen treatment. In both these mouse lines, we detected CD45$^+$ hematopoietic cells tracked by EGFP or ZsGreen fluorescence in BM and blood, which were rare but significantly more abundant than in control mice not treated with tamoxifen (*Figure 7A, B*; *Figure 7—figure supplement 1F–H*). These results indicated that a proportion of the Col1a2-tracked cells in the adult mouse are hemogenic.

To further evaluate this possibility, we first sorted VE-Cadherin$^+$CD45$^-$ ZsGreen (Col1a2)$^+$ cells from the BM of tamoxifen-induced adult Col1a2-Cre^ERT2/ZsGreen mice (*Figure 7—figure supplement 2A*), examined expression of selected genes, and used these cells in transplant experiments. The sorted VE-Cadherin$^+$CD45$^-$ ZsGreen (Col1a2)$^+$ cells expressed *Cdh5*, *Col1a2*, *Cxcl12*, and *Runx1* mRNAs distinctively from other BM cell populations (*Figure 7—figure supplement 2B*), but resembled the BM hemogenic population annotated as 'Mesenchymal-type' (cluster 14) identified by Polylox s.c. sequencing (*Figure 5G*; *Figure 5—figure supplement 1E*) and subsets of adult BM Cdh5$^+$ cells identified in public adult datasets (*Figure 6E, F*).

We transplanted the BM VE-Cadherin$^+$CD45$^-$ ZsGreen (Col1a2)$^+$ cells (1 × 10$^4$ cells/mouse) into 5-FU-conditioned adult WT C57Bl/6 recipients and looked for tracked CD45$^+$ cells in the BM and blood (*Figure 7C*). Four weeks after transplantation, BM and peripheral blood of transplant recipients contained ZsGreen$^+$CD45$^+$ cells (*Figure 7D*; *Figure 7—figure supplement 2C, D*), indicating that the transplanted ZsGreen$^+$ (Col1a2)-tracked CD45$^-$ ECs had produced CD45$^+$ hematopoietic cell progeny. These ZsGreen$^+$CD45$^+$ cells in the recipient mice comprised mainly granulocytes and monocytes, and few B and T lymphocytes (*Figure 7E*). These results indicate that Col1a2-tracked ECs, like Cdh5(VE-Cadherin)-tracked ECs, can give rise to hematopoietic progeny in vivo but display a more restricted multilineage potential compared to ECs from Cdh5-Cre^ERT2(PAC)/ZsGreen mice.

In additional experiments, we evaluated the role of *Runx1* expression in adult BM hemogenic EC since our analyses identified *Runx1* as a putative marker of adult EHT (*Figure 6E, F*; *Figure 5—figure supplement 1E*). Therefore, we generated a Runx1^EC-KI mouse line (Cdh5-Cre^ERT2/ZsGreen/Runx1-Knock-in), in which Cre-mediated excision of a floxed STOP codon enables co-expression of ZsGreen and *Runx1* in ECs upon tamoxifen induction (*Figure 7—figure supplement 2E*; *Qi et al., 2017*). In these Runx1^EC-KI mice, tamoxifen treatment significantly increased the frequency of ZsGreen$^+$ cells in PBMC over 60 weeks compared to control tamoxifen-treated (Cdh5-Cre^ERT2/ZsGreen) mice (*Figure 7F*).

Since these circulating ZsGreen-tracked cells presumably represent hematopoietic cell progeny of ECs co-expressing ZsGreen and *Runx1*, we tested the hemogenic potential of these tamoxifen-induced BM EC ex vivo. Using the culture system that successfully supported ex vivo hematopoiesis in Cdh5-tracked BM (*Figure 2A–H*), we now compared the hemogenic potential of BM cells from tamoxifen pretreated Runx1^EC-KI mice to that of BM cells from tamoxifen pretreated Cdh5-Cre^ERT2/ZsGreen mice. We detected a significantly greater number of ZsGreen$^+$ clusters in cultures from Runx-1^EC-KI BM cells compared to control BM cells (*Figure 7G, H*), and flow cytometry showed that more CD45$^+$ZsGreen$^+$ hematopoietic cells were produced in cultures of Runx1^EC-KI BM cells compared to

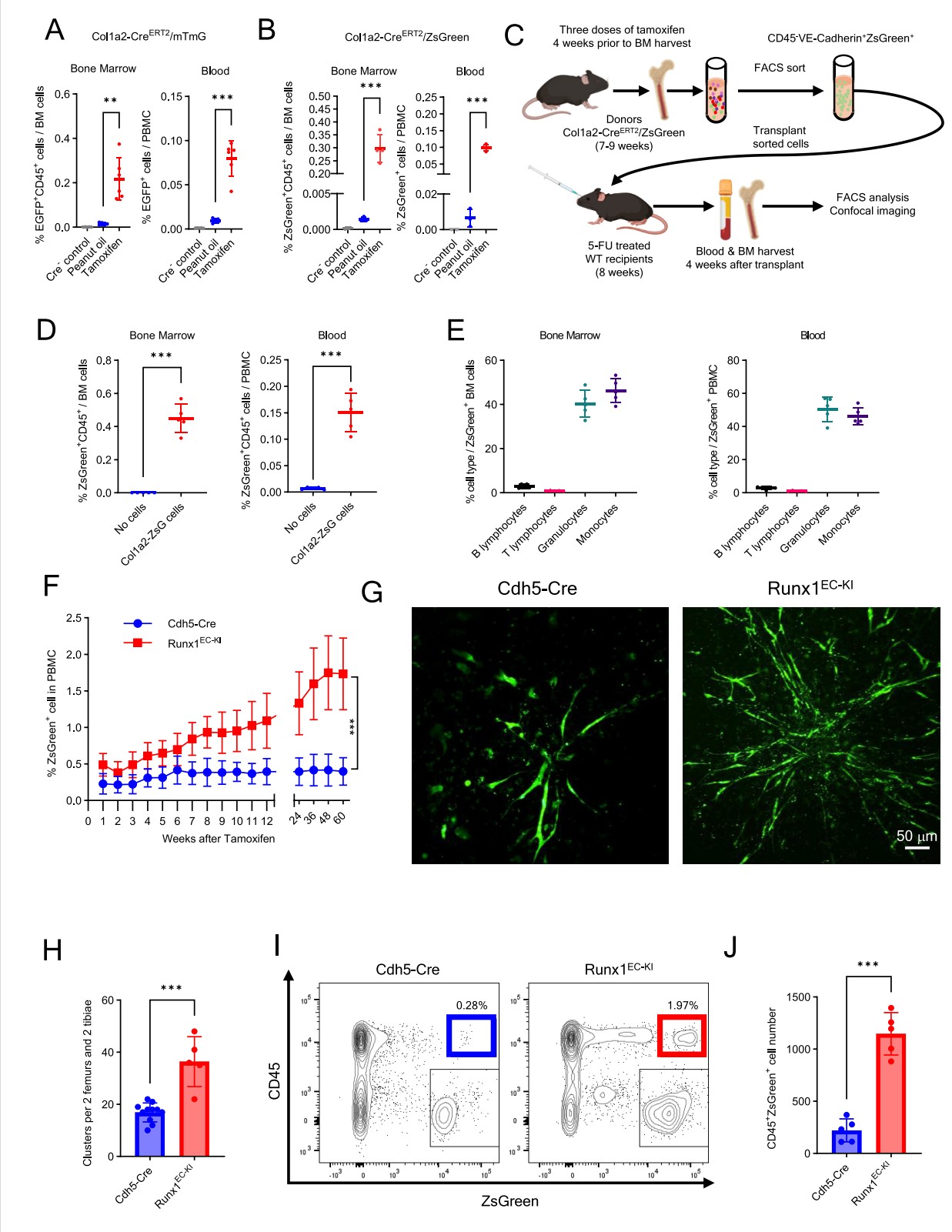

**Figure 7.** Contribution of *Col1a2* and *Runx1* expression to endothelial cells (ECs) hemogenic activity. Percent EGFP+CD45+ cells in bone marrow (BM) and blood of tamoxifen-treated (*n* = 6) or oil-treated (*n* = 5) Col1a2-CreERT2/mTmG mice (**A**) and tamoxifen-treated (*n* = 4) or oil-treated (*n* = 3) Col1a2-CreERT2/ZsGreen mice (**B**). Cre-control mice (*n* = 5 in A, and *n* = 2 in B). (**C**) Transplant experiment: sorted VE-Cadherin+CD45−ZsGreen+/Col1a2+ cells from tamoxifen-treated Col1a2-CreERT2/ZsGreen mice are transplanted into 5-FU-conditioned WT recipients. Detection (**D**) and characterization

*Figure 7 continued on next page*

*Figure 7 continued*

(**E**) of ZsGreen$^+$CD45$^+$ cells in BM and blood of WT 5-FU-conditioned mice (*n* = 5), 4 weeks post-transplant of VE-Cadherin$^+$CD45$^-$ZsGreen$^+$/Col1a2$^+$ cells. Control FU-conditioned WT mice (*n* = 4) received no cell transplant (**D**). (**F**) Time course of ZsGreen$^+$ peripheral blood mononuclear cell (PBMC) detection in control (Cdh5-Cre$^+$/ZsGreen$^+$) and Runx1$^{EC-KI}$ (Cdh5-Cre$^+$/ZsGreen$^+$/Runx1-KI) mice (*n* = 10 per group). Representative images (**G**) and quantification (**H**) of ZsGreen$^+$ cells from OP9 cell-supported cultures of BM cells from tamoxifen-treated Cdh5-Cre$^+$/ZsGreen$^+$ (*n* = 11) and Runx1$^{EC-KI}$ mice (*n* = 5). Representative flow cytometry plots (**I**) and quantification (**J**) of CD45$^+$ZsGreen$^+$ cells from OP9 cell-supported BM cell cultures (*n* = 5/ group). Dots represent individual mice. Data are shown as mean ± SD. **p < 0.01, ***p < 0.001 by Student's *t*-test.

The online version of this article includes the following figure supplement(s) for figure 7:

**Figure supplement 1.** Characterization of Col1a2-tracked cell populations in bone marrow (BM) and blood.

**Figure supplement 2.** Analysis and hemogenic potential of Col1a2-tracked adult bone marrow (BM) endothelial cells (ECs).

BM from controls (*Figure 7I, J*). Although we cannot exclude the possibility that RUNX1 promotes EC proliferation in culture, these results support a role of RUNX1 in promoting adult EHT.

## Discussion

Our results provide evidence for the presence of ECs in the adult mouse BM with hemogenic potential. Previously, hemogenic ECs were detected during embryonic development or perinatally but not thereafter (*Jaffredo et al., 1998*; *Zovein et al., 2008*; *Bertrand et al., 2010*; *Boisset et al., 2010*; *Yokomizo and Dzierzak, 2010*; *Chen et al., 2011*; *Frame et al., 2016*; *Rhodes et al., 2008*; *Nakano et al., 2013*; *Yvernogeau et al., 2019*). The current findings argue that EHT is not limited to the prenatal or perinatal development but is present up to 28 weeks after birth, decreasing thereafter. This conclusion is supported by Cdh5-based bulk and Polylox sc lineage tracking, culture of hemogenic ECs, transplant analyses and initial characterization of EC-derived hematopoietic cell progeny, which link features of adult EHT to embryonic EHT (*North et al., 1999*; *Chen et al., 2009*; *Eliades et al., 2016*; *Yzaguirre et al., 2018*; *Richard et al., 2013*).

Previous experiments found that EHT, present in the late fetus/neonatal BM, disappears shortly after birth (*Yvernogeau et al., 2019*). This contrasts with the current experiments showing persistence of adult EHT well beyond 20 weeks of age. The divergent results likely stem from functional differences of the Cdh5-based tracking systems. In the previous experiments, attempts to activate Cdh5-fluorescence in BM ECs by injecting tamoxifen at different time points after birth were unsuccessful starting 20 days after birth. Expectedly, the absence of tamoxifen-induced fluorescence in BM ECs was associated with the absence of traced hematopoietic cell output from these cells. In the current experiments and those of others (*Wang et al., 2013*), tamoxifen administration after 10, 20, and 30 weeks of age consistently induces fluorescence in most BM ECs. We conclude that the absence of adult EHT had not been firmly established.

Adult hemogenic ECs identified here express *Cdh5*, *Pecam1*, *Kdr*, and *Flt1*, resembling other BM EC populations and embryonic hemogenic ECs (*Dignum et al., 2021*). However, the current studies identify yet another small hemogenic cell population in the adult BM, distinctive in co-expressing typical EC markers, *Cdh5* and *Pecam1*, and the mesenchymal-type markers *Lepr*, *Col1a2*, and *Cxcl12*. These two hemogenic populations are clonally linked, but their relationship is currently unclear.

ECs derive from two sources during development: the splanchnic mesoderm that gives rise to the primitive aorta where ECs located on the aortic floor are hemogenic (*Richard et al., 2013*), and the somites from the paraxial mesoderm, that produce ECs contributing to the endothelial vascular network of the trunk and limb (*Ambler et al., 2001*; *Pudliszewski and Pardanaud, 2005*). Somite-derived ECs are not hemogenic in situ, but they can transiently acquire a hemogenic potential when variously induced (*Pardanaud and Dieterlen-Lièvre, 1999*) and may also include a cell subset with hemogenic potential (*Qiu et al., 2016*). They can also express *Runx1* and trigger aortic hematopoiesis from hemogenic ECs in zebrafish (*Nguyen et al., 2014*) and in the chick (*Richard et al., 2013*). Besides generating hemogenic ECs, blood and other tissues, the mesoderm in conjunction with the neural crest gives rise to mesenchymal stem cells (MSCs) and perhaps a precursor of both MSCs and ECs (*Morikawa et al., 2009*; *Takashima et al., 2007*; *Vodyanik et al., 2010*). However, MSCs do not produce blood cells, although they can differentiate into many other tissues (*Bianco et al., 2008*), and their potential for endothelial differentiation remains controversial (*Oswald et al., 2004*; *Conrad*

*et al., 2009*; *Ubil et al., 2014*; *Crisan, 2013*). By contrast, endothelial to mesenchymal transition (EndMT) is a well-established process in development and disease states (*Xu and Kovacic, 2023*).

It was suggested that the transient wave of EHT occurring in the BM of the late fetus and perinatally may serve to mitigate the slow HSC colonization of BM from the fetal liver or perhaps prepare the BM niches to accommodate the incoming HSC from the liver, but a function has not yet been firmly established (*Yvernogeau et al., 2019*). Similarly, the function of adult BM EHT identified here is currently unclear. We hypothesize that adult EHT plays a functional role under conditions of hematopoietic stress or disease rather than under steady-state conditions, but this will need future investigation. Interestingly, previous studies found that subsets of ECs can regenerate after irradiation that has eliminated HSC (*Xu et al., 2018*), and we traced the emergence of some hematopoietic cells from putative ECs that persisted after lethal mouse irradiation. We also found that 5-FU treatment was required for the successful engraftment and function of adult hemogenic ECs, suggesting that this population may require inductive signals from a BM that is recovering from an insult (*Xu et al., 2018*; *Termini et al., 2021*). The identification of such inductive factors may enable effective propagation of BM hemogenic ECs ex vivo and motivate a search for hemogenic ECs in human BM.

Altogether, our results point to a previously unrecognized capability of ECs in the adult mouse BM to generate blood cells. These results suggest that hematopoiesis in the adult mouse may arise through the contribution of cells and processes beyond the HSCs generated through aortic EHT during development and BM EHT perinatally.

## Limitations of the study

A limitation of our study is that the function of adult EHT is currently unknown. Under steady-state conditions, the hematopoietic cell output from adult EHT is very low in comparison to that of embryonic EHT, which effectively sustains adult hematopoiesis, and our experiments detected no functional differences between hematopoietic cells from embryonic and adult EHT. To address this limitation, we will examine whether adult EHT is a greater contributor to adult hematopoiesis when stress is imposed on the adult hematopoietic system. In addition, we have not fully characterized the hematopoietic cell population(s) arising from adult EHT or studied the time-course kinetics of hematopoietic cell emergence. A key issue is whether long-term repopulating HSPCs are produced by adult EHT. This will require serial transplantation experiments, including CD45.1/CD45.2 competitive repopulation assays, and kinetic analysis of their emergence from hemogenic ECs. Finally, we could not extend the in vivo single-cell PolyloxExpress genetic lineage tracing to the ex vivo single-cell genetic lineage tracing due to technical limitations of the PolyloxExpress system.

## Resource availability

### Lead contact

Further information and requests for resources and reagents should be directed to and will be fulfilled by the lead contact, Giovanna Tosato (tosatog@mail.nih.gov).

### Materials availability

This study did not generate new unique reagents.

# Materials and methods
## Experimental model and study participant details
### Mouse strains

All animal studies were approved by the Institutional Animal Care and Use Committee (IACUC) of the CCR (Bethesda, MD), National Cancer Institute (NCI), NIH and were conducted in strict adherence to the NIH Guide for the Care and Use of Laboratory Animals (National Academies Press, 2011) and study approved protocols (MB-068, MB-088, and POB-027). Cdh5-Cre^ERT2^(PAC) mice (*Sörensen et al., 2009*) (MGI:3848982) were a gift from Dr. R. Adams and Cdh5-Cre^ERT2^(BAC) mice (*Okabe et al., 2014*) (MGI:5705396) were a gift from Dr. Kubota. *Col1a2-CreER* mice (*Zheng et al., 2002*) were purchased from the Jackson Laboratory (JAX#029567). Cre-dependent Ai6 (RCL-ZsGreen) (*Madisen et al., 2010*) (JAX#007906) and mTmG (*Muzumdar et al., 2007*) (JAX#007676) fluorescent reporter mice were purchased from the Jackson Laboratory. In *ZsGreen* reporter mice, Cre activity leads to

constitutive expression of ZsGreen1 in cell bodies. In *mTmG* reporter mice, Cre activity leads to an irreversible switch from cell membrane-localized tdTomato (mT) to membrane-localized EGFP (mG). PolyloxExpress mice (*Pei et al., 2020*) were a kind gift of Drs. Hans-Reimer Rodewald and Avinash Bhandoola. The Cdh5-Cre^ERT2/ZsGreen/PolyloxExpress mouse line was generated in-house. Runx1 conditional knock-in mice (*Qi et al., 2017*) (Gt(ROSA)26Sor^tm1(CAG-Runx1)Lzjg, MGI:7490340) were generously provided by Drs. Qiufu Ma and Nancy Speck. Runx1 endothelial-specific knock-in (*Runx1*^EC-KI) mice were generated by crossing *Runx1*^Ki/+ mice with Cdh5-Cre^ERT2(PAC)/ZsGreen^Tg/Tg mice. Upon tamoxifen treatment, Cre-mediated excision of the two floxed STOP codons enables co-expression of Runx1 and ZsGreen in ECs.

All animals were bred in the animal facilities of CCR/NCI (Bethesda, MD). The mice were maintained in a C57BL/6J background. Mice were identified with ear tags and routinely genotyped by PCR. No mouse was excluded from the experiments, unless assessed as sick by the veterinarians or fight wounds were observed at harvest. Tamoxifen (Sigma-Aldrich, #T5648) dissolved in peanut oil (Sigma-Aldrich, #P2144) (10 mg ml⁻¹) was administered orally (via gavage using 22 g feeding needles) at 100 mg kg⁻¹. Unless otherwise specified, three doses on consecutive days were administered. Unless indicated otherwise, 8- to 12-week-old male and female mice were used when tamoxifen was administrated. No randomization or blinding was used to allocate experimental groups. Mice were typically sacrificed between 9 AM and 11 AM local time.

## Method details

### OP9 cell culture

OP9 cells, a gift from *Nakano, 1996* (also available from ATCC, #CRL-2749), were maintained in α-MEM (Gibco #12561056; without ribonucleosides and deoxyribonucleosides, with 2.2 g/l sodium bicarbonate) supplemented with 20% fetal bovine serum (FBS; Sigma-Aldrich #2442). The cell line was not authenticated prior to use in our study. The cell line tested negative for mycoplasma. Culture dishes and flasks (Corning #353003, #430641U) were precoated with gelatin (Sigma-Aldrich #G9391; ~100 μl/cm², 60 min at 37°C). Cells were incubated at 37°C in a humidified atmosphere of 95% air and 5% $CO_2$.

### BM cell harvesting, culture, and terminal harvest for analysis

To harvest BM cells to be cultured, mice were euthanized by cervical dislocation and soaked in 70% ethanol for 5 min. Long bones (femurs and tibiae) were dissected, and surrounding skin, muscle, and connective tissue were carefully removed. Cleaned bones were immediately transferred into ice-cold sterile PBS and kept on ice. Once all bones were harvested, they were soaked in 70% ethanol for 1 min and rinsed three times with ice-cold PBS.

Each bone was cut into two pieces using sterile scissors and placed—with the open end facing downward—into a sterile 500 μl microcentrifuge tube pre-perforated with a 16 G needle (sterilized in advance). Each 500 μl tube was loaded with two femurs and two tibiae. To each tube, 200 μl of DMEM supplemented with 1 mM EDTA was added. The 500 μl tubes were then placed inside sterile 1.5 ml microcentrifuge tubes, sealed with Parafilm, and centrifuged at 12,000 rpm for 20 s. The inner tubes were discarded, and the BM pellet collected in the 1.5 ml tubes was resuspended in 1 ml DMEM + 1 mM EDTA.

Cells were filtered through 70 μm cell strainers, centrifuged at $350 \times g$ for 5 min at 4°C, and resuspended in complete DMEM (DMEM containing 15% FBS (Sigma-Aldrich #2442-500ML, LOT:24G002), Penicillin-Streptomycin [Gibco #15140-122], and Anti-Anti [Gibco #15240-062]). A single-cell suspension was prepared by pipetting 30 times with a 1-ml pipette.

BM cells were cultured either on Corning Primaria dishes/flasks (Corning #353808, #353810, and #353846) or on OP9 stromal cell monolayers in standard tissue culture dishes/flasks (Corning #353003 and #430641U) precoated with gelatin (Sigma-Aldrich #G9391; ~100 μl/cm², incubated 60 min at 37°C).

Cells were seeded at a density equivalent to BM cells from two femurs and two tibiae per ~75 cm² culture surface (e.g., T75 flask) in 15 ml complete DMEM. After ~32 hr, non-adherent cells and medium were removed, and adherent cells were gently washed three times with PBS. For Primaria cultures, 15 ml fresh complete DMEM was replenished twice weekly. For OP9 co-cultures, non-adherent cells and medium were removed twice weekly and replaced with fresh complete DMEM supplemented

with freshly isolated unfractionated WT BM cells (from two femurs and two tibiae per ~75 cm² surface, using the same BM isolation method mentioned above).

For terminal collection, at the start of the final culture week, all non-adherent cells were removed, and adherent cells were washed three times with PBS. Fresh complete DMEM (15 ml) was added. After 3 days, a second 15-ml complete DMEM addition was made. No further medium changes occurred before harvest. For the final collection, non-adherent cells and supernatant were collected first. Adherent cells were washed with PBS, and the wash was pooled with the supernatant. Remaining adherent cells (including OP9 and BM-derived cells) were detached using 0.25% Trypsin-EDTA (Gibco #25200-056) for 5 min at 37°C and added to the pooled suspension. After another PBS wash, a second trypsinization (10 min at 37°C) was performed. All collected material from each step was combined for downstream flow cytometric analysis. For transplantation experiments, only non-adherent cells (with supernatant) and loosely adherent cells recovered after the initial 5-min Trypsin-EDTA incubation were pooled and subjected to FACS sorting to isolate ZsGreen⁺ cells for injection.

### Blood collection

For terminal collection, blood was obtained from the mouse abdominal aorta with BD Vacutainer EDTA tubes (BD #367856) and Vaculet blood collection needles (23G, EXELINT #26766). Blood smears were prepared with 10 µl of collected blood. For flow cytometry analysis, ACK buffer (Lonza, #BP10-548E) was added to the blood to lyse red blood cells before Fc receptor blocking and antibody staining. For non-terminal blood collection, ~20–50 µl blood was collected by submandibular blood sampling, using a 3-mm animal lancet (BRAINTREE SCI., GR-3MM) and a 250-µl BD Microtainer K2EDTA tube (BD #365974). Kwik Stop Styptic Powder was applied to stop the bleeding (Miracle Corp #423615). For long-term, repeated blood collection, two drops of blood were collected from the tail vein with Microhematocrit Capillary Tubes (Fisherbrand # 22-362574). WBC counts were determined using acridine orange/propidium iodide (AO/PI, Logos Biosystems, #F23001) staining and quantified with a LUNA-FL fluorescence cell counter (Logos Biosystems). For flow cytometry detection of ZsGreen/EGFP positive PBMCs, blood was collected with one of the methods above. For flow cytometric detection of ZsGreen⁺/EGFP⁺ PBMCs, blood was collected using one of the three methods described above. Red blood cells were lysed with ACK buffer, and DAPI and DRAQ5 were used to exclude dead cells and to identify nucleated PBMCs.

### BM harvest for flow cytometry analysis

For flow cytometry analysis, BM was harvested using one of the two methods described below. For hematopoietic cell isolation, BM was harvested by flushing femurs and tibiae with ice-cold Sort Buffer (1× PBS [Gibco, #10010-031] supplemented with 5 mM EDTA, 25 mM HEPES, and 2% FBS [Sigma-Aldrich, #F2442]). Red blood cells were then lysed using ACK lysing buffer (Lonza, #10-548E) according to the manufacturer's instructions. Cells were then washed with Sort Buffer and passed through a 40-µm cell strainer (GREINER BIO-ONE, #542040 and #542140). For greater preservation of ECs, BMs were harvested by gently crushing mouse femurs and tibiae in Sort Buffer (5 mM EDTA, 25 mM HEPES, 2% FBS in 1× PBS). Red cell lysis was performed using ACK buffer. BM cells were then incubated with 0.1 U ml⁻¹ Collagenase (Worthington Biomedical Corp., #LS004176), 0.8 U ml⁻¹ Dispase (Worthington Biomedical Corp., #LS02109), and 0.5 mg ml⁻¹ DNase (Worthington Biomedical Corp., #LS006344) in 1x Hanks' Balanced Salt Solution (HBSS) with $Ca^{2+}$ and $Mg^{2+}$ (Gibco, # 14065056) at 37°C for 30 min on a rotating mixer. Cells were then washed with Sort Buffer and passed through a 40-µm cell strainer.

### EC transplantation experiments

Four days prior to transplant, recipient mice received one dose of 5-FU (150 mg kg⁻¹ in PBS, Sigma-Aldrich, #F6627) intraperitoneally under isoflurane anesthesia. Sorted BM cells (Col1a2-Cre/ZsGreen: 5000 cells; Cdh5-Cre/ZsGreen: 20,000 cells in 100 µl PBS) were inoculated retro-orbitally under isoflurane anesthesia. BMs and blood were harvested from transplant recipients 4 weeks after the transplant unless otherwise specified.

## BM ablation and transplantation

Recipient mice were lethally irradiated (11 Gy) using a Cesium-137 ($^{137}$Cs) gamma irradiator 3 days prior to transplantation.

For whole BM cell transplantation into *mTmG* recipients, BM cells were harvested from WT donor mice by flushing (as described above); $5 \times 10^6$ cells were transplanted per recipient via tail vein. No rescue BM cells were infused.

For LSK transplantation into WT recipients, BM from Cdh5-Cre$^{ERT2}$(PAC)/ZsGreen mice (not treated with tamoxifen) was harvested by flushing, followed by red blood cell lysis with ACK buffer. Fc receptor blocking was performed using Azide-Free Fc Receptor Blocker (Innovex, #NB335-60) per the manufacturer's instructions. Cells were stained with antibodies to Lineage cocktail, Sca-1, and c-Kit. DAPI was used to exclude dead cells. LSK populations were sorted using Sony SH800S and its Ultra Purity mode. Transplanted cell numbers were as follows: ZsGreen$^-$ LSKs ($5 \times 10^4$, *n* = 3; $2.5 \times 10^4$, *n* = 3) and ZsGreen$^+$-enriched LSKs (2800 cells, *n* = 2). No rescue BM cells were infused.

For transplantation of ex vivo-cultured BM cells, non-adherent and loosely adherent ZsGreen$^+$ cells were collected from 8-week OP9 co-cultures (as described above), sorted by FACS for live (DAPI$^-$) ZsGreen$^+$ cells, and transplanted into irradiated recipients at $5 \times 10^4$, $2.5 \times 10^4$, or $1.25 \times 10^4$ cells per mouse (*n* = 2 per group). No rescue BM cells were infused.

## Flow cytometry and cell sorting

For intracellular antigen detection, single-cell suspensions of BM and blood were first incubated with Azide-Free Fc Receptor Blocker (Innovex, #NB335-60), following the manufacturer's instructions. After washing, cells were first stained with surface marker antibodies at the concentration of 2 μg per $1 \times 10^7$ cells in Sort Buffer for 30 min at 4°C and then stained with live/dead cell discriminating BioLegend Zombie Dyes (UV, NIR, Violet, or Yellow, BioLegend #423108, #423106, #423114, and # 423104) following the manufacturer's instructions. After washing, cells were fixed in 4% paraformaldehyde for 10 min at 37°C and then permeabilized with 1% saponin (Sigma-Aldrich, # 47036)/Sort Buffer for 30 min on ice. Subsequently, the cells were stained with rat monoclonal primary RUNX1-PE Ab (Invitrogen, # 12-9816-80) in 0.1% saponin/Sort Buffer overnight. After washing and resuspension, propidium iodide (PI, 0.5 μM, Millipore Sigma, # P4170), 7-AAD (1 μg ml$^{-1}$, Millipore Sigma, #A9400) or DAPI (0.5 μg ml$^{-1}$, BioLegend, #422801) was added as a nuclear counterstain. For live cell staining without cell permeabilization, after cell surface antibody staining, cells were washed and suspended in Sort buffer containing PI (0.5 μM, Millipore Sigma, # P4170), 7-AAD (1 μg ml$^{-1}$, Millipore Sigma, #A9400), or DAPI (0.5 μg ml$^{-1}$, BioLegend, #422801), to distinguish dead cells from the live cells. For FACS sorting of live ECs, after Fc receptor blocking, BM cells were first depleted of CD45$^+$ cells with MojoSort mouse CD45 nanobeads (BioLegend #480028) following the manufacturer's protocol and then stained with specific antibodies. For flow cytometry analysis, compensation beads (BD Biosciences, #552844) were used for flow cytometer compensation. Flow cytometric data were acquired with BD FACSCanto-II, BD LSRFortessa, BD FACSymphony A5 (BD Biosciences), Sony SA3800 or Sony ID7000 cell analyzers. Cell sorting was performed with BD FACS Aria III, BD FACS Aria Fusion, or Sony SH800S cell sorters. FSC and SSC profiles were used for excluding dead cells and debris. 7-AAD, PI, DAPI, or BioLegend Zombie Dye was used for excluding dead cells. FSC-W versus FSC-H and SSC-W versus SSC-H were used to gate on single cells. Unless otherwise specified, Fluorescence Minus One (FMO) controls are used for negative gating reference. For BM HSPC analysis, BM cells were harvested followed by lineage-positive cell depletion (BioLegend #480004). Data were analyzed with FlowJo (BD, v10.8.1), SONY ID7000 Software (Version 1.2.0.28212) or FACS Diva (BD, v6.1 and v9.0). FlowJo Plugins UMAP_R (v4.0.4) and FlowSOM (v4.1.0) were used for UMAP dimensional reduction and unsupervised clustering of flow cytometry data. Following sorting, a small aliquot of collected events was re-run on the sorter to assess purity. Analysis was performed using the identical gating hierarchy (FSC/SSC → singlets → live cells → target population).

## BM cryosections

Deeply anesthetized mice were transcardially perfused with 20 ml ice-cold 1x PBS, followed by perfusion with 15 ml ice-cold hydrogel solution (5% acrylamide/bis-acrylamide 19:1 (Sigma-Aldrich, #A2917), 2.5 mg/ml polymerization initiator VA-044 (FUJIFILM, Wako, VA-044, Water soluble Azo initiators), 4% PFA in 1× PBS, 5 ml/min flow speed). Femurs and tibiae were collected in tubes containing

5 ml hydrogel solution and incubated at 4°C for 4 hr. The bones were then washed with PBS and incubated at 37°C for 2 hr. Bone decalcification was performed by incubating the bones in 40 ml 0.5 M EDTA pH 8.0 (KD Medical, #RGF-3130) for 3 days on a rotate mixer, with daily refreshed 0.5 M EDTA solution. The bones were then dehydrated in 20% sucrose and 2% polyvinylpyrrolidone in PBS overnight. Dehydrated bones were then embedded in OCT (SAKURA, #4583) blocks using Precision Cryoembedding System (IHC WORLD, #IW-P101). Cryosections (10 µm) for immunofluorescence staining were prepared from OCT frozen bone blocks using Leica CM3050S microtome, low-profile microtome blades (Leica 819, #14035838925), and TruBond 380 adhesion slides (Electron Microscopy Sciences, # 63701-W10).

## Immunofluorescence staining and imaging

Tissue sections were rehydrated with 1× PBS (15 min), permeabilized in 0.3% Triton X-100 (Sigma-Aldrich, #T9284)/PBS (15 min), washed in 1×PBS, and incubated (2 hr) with blocking solution (2% BSA, 5% donkey serum (Sigma, #D9663), and 0.3% Triton X-100/PBS). Samples were then washed three times with PBS and incubated with primary antibodies (5 ng/ml; 4°C overnight). When secondary antibodies were used, three PBS washes were performed before incubating with fluorescent-labeled secondary antibodies (2 ng/ml, room temperature, 2 hr). After washing (3x, 10 min each with 1× PBS), DAPI was added (300 nM in PBS, 10 min). After three washes (5 min each with 1× PBS), coverslips were mounted (EPREDIA, #9990402), dried, and sealed with nail polish. For blood smear staining, slides were first soaked in acetone/methanol/PFA (19:19:2 for 90 s) before rehydration (*Happerfield et al., 2008*). Confocal imaging was performed with Zeiss LSM 780, Zeiss LSM 880 NLO Two Photon, or Nikon *ECLIPSE* Ti2-E SoRa systems, according to the experimental specific needs (resolution, speed, wavelength capabilities). Images were processed with Zen (Zen Black v2.3, release Version 14.0.12.201, Zen Blue Lite v2.5, Carl Zeiss), Bitplane Imaris (v9.7.0, Oxford Instruments), and Photoshop (v23.3.0, Adobe, for whole image contrast and brightness adjustments).

## Isolation of peritoneal cavity cells

To isolate peritoneal cavity cells, mice were euthanized by cervical dislocation, injected intraperitoneally (i.p.) with cold FACS sort buffer (5 ml), massaged, and flicked gently on the abdomen. Peritoneal fluid was then withdrawn slowly and transferred into a polypropylene centrifuge tube on ice prior to centrifugation (350 × *g*, 10 min, at 4°C) and analysis.

## TGL-induced sterile peritonitis

Mice were injected i.p. with 1 ml PBS or 4% TGL (Sigma-Aldrich, #70157). Peritoneal cavity cells were harvested 4 hr after injection for further analysis.

## Phagocytosis assay

Phagocytosis assay was performed using pHrodo Red *E. coli* BioParticles (Invitrogen, # P35361) following the manufacturer's instructions. Briefly, mice were first treated with TGL (as described). Peritoneal cavity cells were incubated with the fluorescent *E. coli* particles for 60 min at 37°C. Cells were then stained with surface marker antibodies and analyzed by flow cytometry.

## ROS assay

ROS assay was performed using CellROX Orange Flow Cytometry Assay Kit (Invitrogen, # C10493) following the manufacturer's instructions. Briefly, mice were first treated with TGL (as described). CellROX detection reagent was added to peritoneal cavity cells (final concentration of 500 nM) prior to incubation for 60 min at 37°C. Cells were then stained with the surface marker antibodies and analyzed by flow cytometry.

## PolyloxExpress single-cell lineage tracing

These experiments essentially followed published procedures (*Pei et al., 2020*). Briefly, BM cells from tamoxifen-treated Cdh5-Cre/ZsGreen/Polylox mice (treated at 10 weeks old and harvested at 16 weeks old) were enriched for ZsGreen positive cells by FACS. Single-cell capturing was performed with 10x Genomics Chromium Single Cell 3′ Reagent Kits, following the manufacturer's protocols.

After reverse transcription (RT) in droplets, pooled cDNA was amplified and split into two aliquots for parallel transcriptome library preparation and barcode enrichment. Initial library quality control was performed with Agilent TapeStation D5000. For transcriptome analysis, 10 µl (25%) of a 10x cDNA library was fragmented and a gene expression library, generated following protocols in Single Cell 3′ Reagent Kits v3 and v4, was sequenced with Illumina NextSeq 2000 P3/P4 Reagents (28 + 74 bp read length). For Polylox barcode amplification, targeted amplification of barcodes from a 5- to 10-ng aliquot of a 10 x cDNA library was performed by nested PCR. In the first round, primers #2,652 (5′-GCATGGACGAGCTGTACAAG-3′, annealing at the 5′ end of Polylox) and #2,674 (5′-AATGATAC GGCGACCACCGAGATCTACACTCTTTCCCTACACGACGCTC-3′, annealing at the adaptor site (read 1)) were used for amplification for 5 min at 95°C; (30 s at 95°C, 30 s at 57°C, 3 min at 72°C) 12 times; 10 min at 72°C. PCR products purified with 0.7x AMPure beads were used for the second round of PCR using primers #2,426 (5′-CGACGACACTGCCAAAGATTTC-3′, annealing at the 5′ end of Polylox) and #2,676 (5′-AATGATACGGCGACCACCGA-3′, annealing at the 5′ end of primer #2,674), for 5 min at 95°C; (30 s at 95°C, 30 s at 60°C, 3 min at 72°C) 18 times; 10 min at 72°C. The PCR products were purified with AMPure PB beads according to the manufacturer's protocol. Long read amplicon-seq libraries were sequenced by PacBio Sequel II with PacBio Amplicons Library Preparation using SMRT-bell prep kit 3.0. A custom transcriptome reference was built from mouse reference MM10 to include ZsGreen1.

A snakemake workflow with custom python script was used to retrieve cell indexes and Polylox barcodes from the amplicons (https://github.com/CCRSF-IFX/SF_Polylox-BC, copy archived at *Zhao, 2026*). Single-cell transcriptome and single-cell barcodes were linked using the 10x3′ kit cell index and group information. Further analysis and illustrations were generated using Scanpy (https://github.com/scverse/scanpy, *Wolf et al., 2026*). Doublet detection was performed using Scrublet, with the predicted doublet rate calculated based on 10x Genomics guidance about the cell numbers loaded to the microfluidic chips. Batch correction was done using the BBKNN method (https://github.com/Teichlab/bbknn, *Polanski et al., 2023*). Cell types were manually annotated based on canonical marker gene expression, guided by results from three automated annotation tools: (1) decoupleR (https://saezlab.github.io/decoupleR/), using PangLaoDB (https://panglaodb.se/) as reference; (2) scANVI (https://github.com/scverse/scvi-tools, *Gayoso et al., 2026*) using ImmGne (https://www.immgen.org/) as reference; (3) CellTypist, using the embedded Immune_All_Low model. The rare barcodes and their pGen were identified using the MatLab script from the Höefer Lab (https://github.com/hoefer-lab/polylox, *Rößler, 2017*). True barcodes are defined as pGen $<1 \times 10^{-4}$, such that the expectation of a True barcode in detected 4072 cells is 0.407. This work utilized the computational resources of the NIH HPC Biowulf cluster (http://hpc.nih.gov) and Frederick Research Computing Environment (FRCE). Python version: 3.10; R version 4.5.0. Additional details of the Polylox experiment can be found in *Supplementary file 1*.

## Public single-cell RNA-seq data analysis

Raw FASTQ and BAM files were downloaded from publicly available datasets [GSE108885 (*Tikhonova et al., 2019*), GSE108891 (*Tikhonova et al., 2019*), GSE118436 (*Tikhonova et al., 2019*), GSE123078 (*Tikhonova et al., 2019*), GSE122465 (*Baccin et al., 2020*), GSE128423 (*Baryawno et al., 2019*), GSE145477 (*Kalucka et al., 2020*), GSE156635 (*Sivaraj et al., 2021*), GSE137116 (*Zhu et al., 2020*), E-MTAB-8077 (*Kalucka et al., 2020*), GSE230260 (*Lang et al., 2025*), and GSE259382 (*Smith et al., 2025*)]. The BAM files were first converted to FASTQ files with bamtofastq (10x Genomics, v2.0.1). All FASTQ files were then processed by the count function of Cell Ranger (10x Genomics, v8.0.1) and aligned to the mouse genome (mm10, version 2020-A), to generate read matrices. Further analysis and illustrations were generated using Scanpy as described above.

## Definition of cultured fluorescent BM cell clusters and quantification

A cluster was defined as (1) a spatially distinct group of ZsGreen$^+$ cells not contiguous with other fluorescent cells, (2) containing a central core with at least five extending branches, and (3) exhibiting a roughly radial organization, with projections spreading outward in multiple directions, and (4) extending at least ~200 µm in one direction. Isolated or irregularly scattered fluorescent cells were not counted. Due to their large size, clusters were counted manually across the culture dish using a tally counter under fluorescence microscopy.

## RNA isolation and quantitative RT-PCR

RNA was extracted using RNeasy Micro Kit (QIAGEN, #74004), following the manufacturer's protocol. Cells were sorted directly into RNA lysis buffer (Buffer RTL of RNeasy Micro Kit). cDNA samples were prepared with SuperScript IV Reverse Transcriptase (Invitrogen, #18091050), following the manufacturer's instructions. Real-Time PCR was performed using Applied Biosystems TaqMan Fast Advanced Master Mix (4444557) and Applied Biosystem QuantStudio 5 Real-Time PCR System. TaqMan probes used were purchased from Applied Biosystems: Spp1 (Mm00436767_m1); Cxcl12 (Mm00445553_m1); Col1a2 (Mm00483888_m1); Cdh5 (Mm00486938_m1); Runx1 (Mm01213404_m1); Ptprc (Mm01293577_m1); Gapdh (Mm99999915_g1); Actb (Mm02619580_g1). The reaction condition was set as follows: 50°C 2 min, 95°C 20 s, 45 cycles of 95°C 1 s, 60 °C 20 s. Ct values were determined using the ABI QuantStudio Design & Analysis Software (v1.5.2). Relative gene expression was assessed using the $2^{-\Delta\Delta Ct}$ method, normalized to Gapdh expression level for each sample. The data was further normalized to gene expression levels in the unsorted BM sample to calculate relative gene expression levels in each sample. Data reflect triplicates real-time PCR experiments.

## Quantification and statistical analysis

No statistical method was used to predetermine sample size. No data were excluded from the analyses. Mice with the correct genotypes were randomly assigned to control or treated groups. Unless otherwise specified, data are represented as mean ±SD, and individual dots in the graphs indicate individual mice. Comparisons between two groups were performed using two-tailed unpaired Student's *t*-tests (except for *Figure 1—figure supplement 1J*, which used a paired *t*-test). Spearman rank correlation test was used for *Figure 3E*. Statistical analyses were performed with GraphPad Prism (v9.0.1). A statistical difference of $p < 0.05$ was considered significant: ns, not significant, *$p < 0.05$, **$p < 0.01$, ***$p < 0.001$.

## Acknowledgements

This project is supported by the Intramural Program of CCR, NCI, and NIH. The findings are those of the authors and do not necessarily represent the official views of the NIH or the Department of Health and Human Services. This work used the computational resources of the NIH High Performance Computing (HPC) Biowulf cluster (https://hpc.nih.gov) and Frederick Research Computing Environment (FRCE). Flow cytometry and cell sorting were performed at the CCR Flow Cytometry Core Facility; microscopy analyses at the CCR Microscopy Core Facility in Building 37 of the NCI, NIH. We thank Drs. Ralf Adams and Manfred Boehm for mouse line Cdh5-Cre[ERT2](PAC); Drs. Yoshiaki Kubota and Yosuke Mukoyama for mouse line Cdh5-Cre[ERT2](BAC); Dr. Hans-Reimer Rodewald for the PolyloxExpress mouse line. We thank Dr. S Banerjee, Dr. S Siddiqui, and Ms. K M Wolcott for flow cytometry support; Dr. M Kruhlak and Mr. L Lim for confocal microscopy support and Mr. A Abdelmaksoud for bioinformatics assistance. We thank Drs. D Lowy, R Yarchoan, M DiPrima, and H Ohnuki for thoughtfully commenting on aspects of this work.

---

# Additional information

### Funding

| Funder | Grant reference number | Author |
|---|---|---|
| CCR/NCI/NIH | ZO1 SC 010355 | Giovanna Tosato |

The funders had no role in study design, data collection, and interpretation, or the decision to submit the work for publication.

### Author contributions

Jing-Xin Feng, Conceptualization, Data curation, Software, Formal analysis, Validation, Investigation, Methodology, Writing – review and editing; Mei-Ting Yang, Data curation; Lili Li, Formal analysis, Methodology; Caiyi C Li, Dunrui Wang, Formal analysis; Ferenc Livak, Formal analysis, Validation; Jack Chen, Software, Validation; Yongmei Zhao, Data curation, Formal analysis; Avinash Bhandoola, Naomi

Taylor, Resources; Giovanna Tosato, Conceptualization, Resources, Supervision, Funding acquisition, Investigation, Visualization, Methodology, Writing – original draft, Project administration, Writing – review and editing

**Author ORCIDs**
Jing-Xin Feng ⓘ https://orcid.org/0000-0003-0677-2792
Mei-Ting Yang ⓘ https://orcid.org/0000-0003-2820-5940
Dunrui Wang ⓘ https://orcid.org/0000-0003-0310-553X
Giovanna Tosato ⓘ https://orcid.org/0000-0003-1663-3227

**Ethics**
All animal studies were approved by the Institutional Animal Care and Use Committee (IACUC) of the CCR (Bethesda, MD), National Cancer Institute (NCI), NIH and conducted in strict adherence to the NIH Guide for the Care and Use of Laboratory Animals (National Academies Press, 2011) and approved animal protocols (MB-068, MB-088, and POB-027).

Reviewer #1 (Public review): https://doi.org/10.7554/eLife.109553.3.sa1
Reviewer #2 (Public review): https://doi.org/10.7554/eLife.109553.3.sa2
Author response https://doi.org/10.7554/eLife.109553.3.sa3

---

## Additional files

### Supplementary files
MDAR checklist

Supplementary file 1. Distribution of Polylox barcodes among bone marrow cell populations.

### Data availability
The results of scRNAseq and PacBio SmrtSeq of Polylox barcodes are deposited in NCBI SRA (PRJNA1079369). A custom script was used to retrieve cell indexes from the PolyloxExpress amplicons (https://github.com/CCRSF-IFX/SF_Polylox-BC copy archived at *Zhao, 2026*). Bash, R, and Python codes are available at https://github.com/TosatoLab/PolyloxProcessingScripts (copy archived at *Feng, 2026*).

The following dataset was generated:

| Author(s) | Year | Dataset title | Dataset URL | Database and Identifier |
|---|---|---|---|---|
| NIH | 2024 | Polylox tracking of EC in Mouse BM | https://www.ncbi.nlm.nih.gov/bioproject/PRJNA1079369/ | NCBI BioProject, PRJNA1079369 |

The following previously published datasets were used:

| Author(s) | Year | Dataset title | Dataset URL | Database and Identifier |
|---|---|---|---|---|
| Smith Neal P, Janton C, Clara MAA, Majd G, Kasidet M, Guping Mao, Alexandra-Chloé V, Kronenberg Henry M | 2025 | Single cell sequencing shows that Sox9-expressing cells populate most mesenchymal lineages postnatally in mouse bone | https://www.ncbi.nlm.nih.gov/geo/query/acc.cgi?acc=GSE259382 | NCBI Gene Expression Omnibus, GSE259382 |
| Lang A, Collins JM, Nijsure MP, Belali S | 2025 | Gene expression profile at single cell level of cells isolated from the bone fracture gap and un-injured bone marrow in mouse osteotomy models | https://www.ncbi.nlm.nih.gov/geo/query/acc.cgi?acc=GSE230260 | NCBI Gene Expression Omnibus, GSE230260 |

*Continued on next page*

*Continued*

| Author(s) | Year | Dataset title | Dataset URL | Database and Identifier |
|---|---|---|---|---|
| Khan S | 2020 | A single cell transcriptome atlas of murine endothelial cells | https://www.ebi.ac.uk/biostudies/arrayexpress/studies/E-MTAB-8077 | ArrayExpress, E-MTAB-8077 |
| Zhu Q, Gao P, Tober J, Bennett L | 2020 | Developmental trajectory of pre-hematopoietic stem cell formation from endothelium (scRNA-seq data set) | http://ncbi.nlm.nih.gov/geo/query/acc.cgi?acc=GSE137116 | NCBI Gene Expression Omnibus, GSE137116 |
| Sivaraj KK, Jeong HW, Dharmalingam B, Zeuschner D | 2021 | Regional specialization and fate specification of mesenchymal stromal cells in skeletal development [scRNA-seq] | https://www.ncbi.nlm.nih.gov/geo/query/acc.cgi?acc=GSE156635 | NCBI Gene Expression Omnibus, GSE156635 |
| Zhong L, Yao L, Tower RJ, Wei Y | 2020 | Single cell transcriptomics analysis of bone marrow mesenchymal lineage cells | https://www.ncbi.nlm.nih.gov/geo/query/acc.cgi?acc=GSE145477 | NCBI Gene Expression Omnibus, GSE145477 |
| Baryawno N, Przybylski D, Kowalczyk MS, Kfoury Y | 2019 | A cellular taxonomy of the bone marrow stroma in homeostasis and leukemia demonstrates cancer-crosstalk with stroma to impair normal tissue function | https://www.ncbi.nlm.nih.gov/geo/query/acc.cgi?acc=GSE128423 | NCBI Gene Expression Omnibus, GSE128423 |
| Baccin C, Al-Sabah J, Velten L, Helbling PM | 2019 | Single-cell RNA-seq of frequent and rare populations in mouse bone marrow | https://www.ncbi.nlm.nih.gov/geo/query/acc.cgi?acc=GSE122465 | NCBI Gene Expression Omnibus, GSE122465 |
| Tikhonova AN, Dolgalev I, Hu H, Sivaraj KK | 2019 | Single cell transcriptome profiling of the bone marrow niche at steady state and under stress conditions (validation experiment) | https://www.ncbi.nlm.nih.gov/geo/query/acc.cgi?acc=GSE123078 | NCBI Gene Expression Omnibus, GSE123078 |
| Tikhonova AN, Dolgalev I, Hu H, Sivaraj KK | 2019 | Single cell transcriptome profiling of LSKs in the absence of vascular Dll4 | https://www.ncbi.nlm.nih.gov/geo/query/acc.cgi?acc=GSE118436 | NCBI Gene Expression Omnibus, GSE118436 |
| Tikhonova AN, Dolgalev I, Hu H, Sivaraj KK | 2019 | Single cell transcriptome profiling of the bone marrow niche at steady state and under stress conditions | https://www.ncbi.nlm.nih.gov/geo/query/acc.cgi?acc=GSE108891 | NCBI Gene Expression Omnibus, GSE108891 |
| Tikhonova AN, Dolgalev I, Hu H, Sivaraj KK | 2019 | Gene expression in BM niche populations | https://www.ncbi.nlm.nih.gov/geo/query/acc.cgi?acc=GSE108885 | NCBI Gene Expression Omnibus, GSE108885 |

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

# Appendix 1

### Appendix 1—key resources table

| Reagent type (species) or resource | Designation | Source or reference | Identifiers | Additional information |
|---|---|---|---|---|
| Antibody | BD Horizon BV786 Rat Anti-Mouse CD117 (rat monoclonal, clone 2B8) | BD Biosciences | 564012; RRID:AB_2732005 | Flow cytometry 2 mg/$10^7$ cells |
| Antibody | Brilliant Violet 421 anti-mouse CD117 (c-Kit) Antibody (rat monoclonal, clone 2B8) | BioLegend | 105828; RRID:AB_11204256 | Flow cytometry 2 mg/$10^7$ cells |
| Antibody | Brilliant Violet 785 anti-mouse CD117 (c-Kit) Antibody (rat monoclonal, clone 2B8) | BioLegend | 105841; RRID:AB_2629799 | Flow cytometry 2 mg/$10^7$ cells |
| Antibody | BD Pharmingen APC Rat Anti-CD11b (rat monoclonal, clone M1/70) | BD Biosciences | 553312; RRID:AB_398535 | Flow cytometry 2 mg/$10^7$ cells; immunostaining: 2 ng/ml |
| Antibody | PerCP/Cyanine5.5 anti-mouse CD150 (SLAM) Antibody (rat monoclonal, clone TC15-12F12.2) | BioLegend | 115922; RRID:AB_2303663 | Flow cytometry 2 mg/$10^7$ cells |
| Antibody | APC/Fire 750 anti-mouse CD150 (SLAM) Antibody (rat monoclonal, clone TC15-12F12.2) | BioLegend | 115940; RRID:AB_2629587 | Flow cytometry 2 mg/$10^7$ cells |
| Antibody | BD Pharmingen APC-Cy7 Rat Anti-Mouse CD19 (rat monoclonal, clone 1D3) | BD Biosciences | 557655; RRID:AB_396770 | Flow cytometry 2 mg/$10^7$ cells |
| Antibody | BD Horizon BUV737 Rat Anti-Mouse CD19 (rat monoclonal, clone 1D3) | BD Biosciences | 612781; RRID:AB_2870110 | Flow cytometry 2 mg/$10^7$ cells |
| Antibody | Brilliant Violet 510 anti-mouse CD3 Antibody (rat monoclonal, clone 17A2) | BioLegend | 100234; RRID:AB_2562555 | Flow cytometry 2 mg/$10^7$ cells |
| Antibody | BD Pharmingen Purified Rat Anti-Mouse CD31 (rat monoclonal, clone 390) | BD Biosciences | 553370; RRID:AB_394816 | Flow cytometry 2 mg/$10^7$ cells |
| Antibody | BV421 anti-mouse Cd31 (rat monoclonal, clone 390) | BD Biosciences | 563356; RRID:AB_2738154 | Flow cytometry 2 mg/$10^7$ cells |
| Antibody | Brilliant Violet 421 anti-mouse CD31 Antibody (rat monoclonal, clone 390) | BioLegend | 102424; RRID:AB_2650892 | Flow cytometry 2 mg/$10^7$ cells |
| Antibody | Brilliant Violet 605 anti-mouse CD31 Antibody (rat monoclonal, clone 390) | BioLegend | 102427; RRID:AB_2563982 | Flow cytometry 2 mg/$10^7$ cells |
| Antibody | BD Pharmingen Alexa Fluor 700 Rat Anti-Mouse CD45 (rat monoclonal, clone 30-F11) | BD Biosciences | 560510; RRID:AB_1645208 | Flow cytometry 2 mg/$10^7$ cells; immunostaining: 2 ng/ml |
| Antibody | BD OptiBuild BUV615 Rat Anti-Mouse CD45 (rat monoclonal, clone 30-F11) | BD Biosciences | 751170; RRID:AB_2875194 | Flow cytometry 2 mg/$10^7$ cells |
| Antibody | PE anti-mouse CD45 Antibody (rat monoclonal, clone 30-F11) | BioLegend | 103106; RRID:AB_312971 | Flow cytometry 2 mg/$10^7$ cells; immunostaining: 2 ng/ml |
| Antibody | PerCP/Cyanine5.5 anti-mouse CD45 Antibody (rat monoclonal, clone 30-F11) | BioLegend | 103131; RRID:AB_893344 103132; RRID:AB_893340 | Flow cytometry 2 mg/$10^7$ cells |
| Antibody | BD Horizon BV510 Hamster Anti-Mouse CD48 (Armenian hamster monoclonal, clone HM48-1) | BD Biosciences | 563536; RRID:AB_2738266 | Flow cytometry 2 mg/$10^7$ cells |
| Antibody | APC/Cyanine7 anti-mouse CD48 Antibody (Armenian hamster monoclonal, clone HM48-1) | BioLegend | 103432; RRID:AB_2561463 | Flow cytometry 2 mg/$10^7$ cells |
| Antibody | Brilliant Violet 510 anti-mouse CD48 Antibody (Armenian hamster monoclonal, clone HM48-1) | BioLegend | 103443; RRID:AB_2650826 | Flow cytometry 2 mg/$10^7$ cells |
| Antibody | 'Collagen I Polyclonal Antibody, Biotin' (rabbit polyclonal) | Invitrogen | PA1-28530; RRID:AB_1956957 | Flow cytometry 2 mg/$10^7$ cells |

*Appendix 1 Continued on next page*

*Appendix 1 Continued*

| Reagent type (species) or resource | Designation | Source or reference | Identifiers | Additional information |
|---|---|---|---|---|
| Antibody | 'Endomucin Monoclonal Antibody (eBioV.7C7 (V.7C7)), eFluor 660' (rat monoclonal, clone eBioV.7C7) | Invitrogen | 50-5851-82; RRID:AB_11220465 | Flow cytometry 2 mg/$10^7$ cells; immunostaining: 2 ng/ml |
| Antibody | Anti-Endomucin Antibody (V.7C7) AF546 (rat monoclonal, clone V.7C7) | Santa Cruz Biotechnology | sc-65495 AF546; RRID:AB_2100037 | Flow cytometry 2 mg/$10^7$ cells |
| Antibody | BD Pharmingen PE-Cy7 Rat Anti-Mouse Ly-6G (rat monoclonal, clone 1A8) | BD Biosciences | 560601; RRID:AB_1727562 | Flow cytometry 2 mg/$10^7$ cells |
| Antibody | BD Horizon BUV395 Rat Anti-Mouse Ly-6G (rat monoclonal, clone 1A8) | BD Biosciences | 563978; RRID:AB_2716852 | Flow cytometry 2 mg/$10^7$ cells |
| Antibody | 'BD Pharmingen APC Mouse Lineage Antibody Cocktail, with Isotype Control' (rat/hamster monoclonal cocktail) | BD Biosciences | 558074; RRID:AB_1645213 | Flow cytometry 2 mg/$10^7$ cells |
| Antibody | 'Goat anti-Rat IgG (H+L) Highly Cross-Adsorbed Secondary Antibody, Alexa Fluor Plus 594' (goat polyclonal) | Invitrogen | A48264; RRID:AB_2896333 | Flow cytometry 2 mg/$10^7$ cells |
| Antibody | 'RUNX1 Monoclonal Antibody (RXDMC), PE, eBioscience' (rat monoclonal, clone RXDMC) | Invitrogen | 12-9816-80; RRID:AB_11151519 | Flow cytometry 2 mg/$10^7$ cells |
| Antibody | BD Pharmingen PE-Cy7 Rat Anti-Mouse Ly-6A/E (rat monoclonal, clone D7) | BD Biosciences | 561021; RRID:AB_2034021 | Flow cytometry 2 mg/$10^7$ cells |
| Antibody | PE/Cy7 anti-mouse Ly-6A/E (Sca-1) (rat monoclonal, clone D7) | BioLegend | 108114; RRID:AB_493596 | Flow cytometry 2 mg/$10^7$ cells |
| Antibody | PerCP/Cyanine5.5 anti-mouse TER-119/Erythroid Cells Antibody (rat monoclonal, clone TER-119) | BioLegend | 116228; RRID:AB_893636 | Flow cytometry 2 mg/$10^7$ cells |
| Antibody | Brilliant Violet 605 anti-mouse TER-119/Erythroid Cells Antibody (rat monoclonal, clone TER-119) | BioLegend | 116239; RRID:AB_2562447 | Flow cytometry 2 mg/$10^7$ cells |
| Antibody | BD Pharmingen Alexa Fluor 647 Rat Anti-Mouse CD144 (rat monoclonal, clone 11D4.1) | BD Biosciences | 562242; RRID:AB_2737608 | Flow cytometry 2 mg/$10^7$ cells |
| Antibody | BD Pharmingen PE Rat Anti-Mouse CD144 (rat monoclonal, clone 11D4.1) | BD Biosciences | 562243; RRID:AB_2737609 | Flow cytometry 2 mg/$10^7$ cells |
| Antibody | BUV737 Rat Anti-Mouse CD144 (rat monoclonal, clone 11D4.1) | BD Biosciences | 741792; RRID:AB_2871138 | Flow cytometry 2 mg/$10^7$ cells |
| Antibody | Alexa Fluor 647 anti-mouse CD144 (VE-cadherin) Antibody (rat monoclonal, clone BV13) | BioLegend | 138006; RRID:AB_10569114 | Flow cytometry 2 mg/$10^7$ cells |
| Antibody | BD OptiBuild RB780 Rat Anti-Mouse CD144 (rat monoclonal, clone 11D4.1) | BD Biosciences | 755945; RRID:AB_3683567 | Flow cytometry 2 mg/$10^7$ cells |
| Antibody | PE anti-mouse CD144 (VE-cadherin) Antibody (rat monoclonal, clone BV13) | BioLegend | 138010; RRID:AB_10641139 | Flow cytometry 2 mg/$10^7$ cells |
| Antibody | PE/Cyanine7 anti-mouse CD144 (VE-cadherin) Antibody (rat monoclonal, clone BV13) | BioLegend | 138015; RRID:AB_2562885 138016; RRID:AB_2562886 | Flow cytometry 2 mg/$10^7$ cells |
| Antibody | PE anti-mouse CD144 (VE-cadherin) Antibody (rat monoclonal, clone VECD1) | BioLegend | 138105; RRID:AB_2077941 | Flow cytometry 2 mg/$10^7$ cells |
| Chemical compound, drug | 5-FLUOROURACIL | Sigma-Aldrich | F6627 | |
| Chemical compound, drug | 7-AAD Viability Stain SOLN | Life Technologies Corp | 00-6993-50 | |
| Chemical compound, drug | 7-Aminoactinomycin D | Sigma-Aldrich | A9400 | |
| Chemical compound, drug | ACETONE | MALLINCKRODT | 2440 | |

*Appendix 1 Continued on next page*

*Appendix 1 Continued*

| Reagent type (species) or resource | Designation | Source or reference | Identifiers | Additional information |
|---|---|---|---|---|
| Chemical compound, drug | ACK lysing Buffer | Lonza | 10-548E | |
| Chemical compound, drug | Acrylamide/Bis 19:1, 40% (wt/vol) solution | Invitrogen | AM9024 | |
| Chemical compound, drug | Acrylamide/Bis-acrylamide 19:1 | Sigma-Aldrich | A2917 | |
| Chemical compound, drug | Acridine Orange/Propidium Iodide Stain | Logos Biosystems | F23001 | |
| Chemical compound, drug | AMPure beads | Beckman Coulter | A63881 | |
| Chemical compound, drug | Antibiotic-Antimycotic | Gibco | 15240062 | |
| Chemical compound, drug | Anti-Rat Ig, κ/Negative Control (BSA) Compensation Plus (7.5 µm) Particles Set | BD Biosciences | 560499 | |
| Chemical compound, drug | Anti-Rat Ig, κ/Negative Control Compensation Particles Set | BD Biosciences | 552844 | |
| Chemical compound, drug | APC/Fire 750 Streptavidin | BioLegend | 405250 | |
| Chemical compound, drug | Azide-Free Fc Receptor Blocker | INNOVEX | NB335-60 | |
| Chemical compound, drug | Bovine serum albumin solution | MP Biomedicals | 160069 | |
| Chemical compound, drug | CellROX Orange Flow Cytometry Assay Kit | Invitrogen | C10493 | |
| Chemical compound, drug | Collagenase Type 2 | Worthington Biochemical Corporation | LS004176 | |
| Chemical compound, drug | Dispase | Worthington Biomedical | LS02109 | |
| Chemical compound, drug | Deoxyribonuclease | Worthington Biomedical | LS006344 | |
| Chemical compound, drug | DAPI (4',6-Diamidino-2-Phenylindole, Dilactate) | BioLegend | 422801 | |
| Chemical compound, drug | Dihydroethidium (Hydroethidine) | Invitrogen | D1168 | |
| Chemical compound, drug | Donkey serum | Sigma-Aldrich | D9663 | |
| Chemical compound, drug | DRAQ5 | Biostatus | DR50200 | |
| Chemical compound, drug | EDTA (0.5 M, pH 8.0) | KD Medical | RGF3130 | |
| Chemical compound, drug | ETHYL ALCOHOL (200 PROOF ANHYDROUS) | Warner-Graham Co | 201096 | |
| Chemical compound, drug | Ethylenediamine-$N,N,N',N'$-tetra-2-propanol | Sigma-Aldrich | 8219401000 | |
| Chemical compound, drug | Fetal bovine serum | Millipore Sigma | F2442-500ML (Lot 24G002) | |
| Chemical compound, drug | Fc Receptor Blocker | Innovex | NB309-30 | |
| Chemical compound, drug | Formalin solution, neutral buffered, 10% | Sigma-Aldrich | HT501128-4L | |

*Appendix 1 Continued on next page*

*Appendix 1 Continued*

| Reagent type (species) or resource | Designation | Source or reference | Identifiers | Additional information |
|---|---|---|---|---|
| Chemical compound, drug | Gelatin | Sigma-Aldrich | G9391 | |
| Chemical compound, drug | Goldenrod Animal Lancet | Braintree Scientific Inc | GR-3MM | |
| Chemical compound, drug | HBSS | Gibco | 14025075 | |
| Chemical compound, drug | HEPES | Gibco | 15630080 | |
| Chemical compound, drug | Immu-Mount mountant | Epredia | 9990402 | |
| Chemical compound, drug | Isoflurane | Baxter | 10019036040 | |
| Chemical compound, drug | Mag-Bind TotalPure NGS | Omega Bio-tek | M1378-01 | |
| Chemical compound, drug | Methanol, HPLC Grade | Avantor | JT-9093-03 | |
| Chemical compound, drug | MojoSort Mouse CD45 Nanobeads | BioLegend | 480028 | |
| Chemical compound, drug | Kwik Stop Stypic Power | Miracle Care | SKU 423615 | |
| Chemical compound, drug | Lineage Cell Depletion Kit, mouse | MiltenyiBiotec | 130-090-858 | |
| Chemical compound, drug | Lipopolysaccharides | Sigma-Aldrich | L2880-25MG | |
| Chemical compound, drug | Microscope Slides | MATSUNAMI GLASS IND | SUMGP12 | |
| Chemical compound, drug | Neutral Protease, Partially Purified, Animal Free/AF | Worthington Biochemical Corporation | LS02109 | |
| Chemical compound, drug | Paraformaldehyde (formaldehyde) aqueous solution (20%) | Electron Microscopy Sciences | 15713-S | |
| Chemical compound, drug | Penicillin–streptomycin | Gibco | 15140-122 | |
| Chemical compound, drug | Peanut oil | Sigma-Aldrich | P2144 | |
| Chemical compound, drug | Phusion Green Hot Start II High-Fidelity PCR Master Mix | Thermo Scientific | F566L | |
| Chemical compound, drug | pHrodo Red *E. coli* BioParticles | Invitrogen | P35361 | |
| Chemical compound, drug | Polyvinylpyrrolidone | Sigma-Aldrich | P5288 | |
| Chemical compound, drug | Propidium iodide | Sigma-Aldrich | P4170 | |
| Chemical compound, drug | Q5 Hot Start High-Fidelity 2X Master Mix | NEB | M0494S | |
| Chemical compound, drug | Richard-Allan Scientific Cover glass | Epredia | 102424 | |
| Chemical compound, drug | RNeasy Micro Kit | QIAGEN | 74004 | |

*Appendix 1 Continued on next page*

*Appendix 1 Continued*

| Reagent type (species) or resource | Designation | Source or reference | Identifiers | Additional information |
|---|---|---|---|---|
| Chemical compound, drug | Saponin | Sigma-Aldrich | 47036 | |
| Chemical compound, drug | SPRIselect Beads | Beckman Coulter | B23318 | |
| Chemical compound, drug | Sucrose | Sigma-Aldrich | S8501 | |
| Chemical compound, drug | SuperScript IV First-Strand Synthesis System | Invitrogen | 18091050 | |
| Chemical compound, drug | Tamoxifen | Sigma-Aldrich | T5648 | |
| Chemical compound, drug | TaqMan Fast Advanced Master Mix | Thermo Fisher Scientific | 4444557 | |
| Chemical compound, drug | Thioglycollate Broth | Sigma-Aldrich | 70157 | |
| Chemical compound, drug | Tissue-Tek O.C.T. Compound | SAKURA | 4583 | |
| Chemical compound, drug | Triton X-100 | Sigma-Aldrich | T9284 | |
| Chemical compound, drug | Trypsin-EDTA (0.25%), phenol red | Gibco | 25200056 | |
| Chemical compound, drug | TruBond 380 Adhesion Slide | Electron Microscopy Sciences | 63700-Y10 | |
| Chemical compound, drug | UNI-TRIEVE | INNOVEX | NB325 | |
| Chemical compound, drug | VA-044 (Water soluble Azo initiators) | FUJIFILM Labchem Wako | VA-044 | |
| Chemical compound, drug | VECTASHIELD Vibrance Antifade Mounting Medium | Vector Laboratories | H-1700-10 | |
| Chemical compound, drug | Zombie Aqua Fixable Viability Kit | BioLegend | 423101 | |
| Chemical compound, drug | Zombie NIR Fixable Viability Kit | BioLegend | 423105 | |
| Chemical compound, drug | Zombie UV Fixable Viability Kit | BioLegend | 423107 | |
| Chemical compound, drug | Zombie Yellow Fixable Viability Kit | BioLegend | 423103 | |
| Chemical compound, drug | Zymosan A from *Saccharomyces cerevisiae* | Sigma-Aldrich | Z4250 | |
| Commercial assay or kit | Chromium GEM-X Single Cell 3′ Kit | 10× Genomics | 1000686 | |
| Commercial assay or kit | Chromium GEM-X Single Cell 3′ Chip Kit v4 | 10× Genomics | 1000690 | |
| Commercial assay or kit | Chromium Next GEM Chip G Single Cell Kit | 10× Genomics | 1000127 | |
| Commercial assay or kit | Chromium Next GEM Single Cell 3′ Kit v3.1 | 10× Genomics | 1000269 | |
| Commercial assay or kit | Dual Index Kit TT Set | 10× Genomics | 1000215 | |
| Commercial assay or kit | NextSeq 2000 P4 Reagents (100 Cycles) | Illumina | 20100994 | |

*Appendix 1 Continued on next page*

*Appendix 1 Continued*

| Reagent type (species) or resource | Designation | Source or reference | Identifiers | Additional information |
|---|---|---|---|---|
| Commercial assay or kit | NextSeq 2000 P3 Reagents (100 Cycles) | Illumina | 20040559 | |
| Other | Polylox PacBio long-read sequencing data | This study | PRJNA1079369 | Dataset accession associated with this study; eLife usually requests datasets in the submission metadata rather than in the Key Resources Table. See Materials and methods for details. |
| Other | Single-cell RNA-seq data | This study | PRJNA1079369 | Dataset accession associated with this study; eLife usually requests datasets in the submission metadata rather than in the Key Resources Table. See Materials and methods for details. |
| Cell line (*Mus musculus*) | OP9 cells | Dr. Nakano; same line deposited in ATCC | ATCC # CRL-2749; RRID:CVCL_4398 | Mouse bone marrow stromal cell line. |
| Genetic reagent (*Mus musculus*) | Cdh5-Cre(PAC)ERT2 | Drs. R. Adams and M. Boehm; also available from Taconic Biosciences | MGI:3848982 Taconic # 13073 | Mouse line used in this study. |
| Genetic reagent (*Mus musculus*) | Cdh5-Cre(BAC)ERT2 | Drs. Y. Kubota and Y. Mukoyama | MGI:5705396 | Mouse line used in this study. |
| Genetic reagent (*Mus musculus*) | Col1a2-CreERT | JAX, #029567 | MGI:3785760 | Mouse line used in this study. |
| Genetic reagent (*Mus musculus*) | ZsGreen (Ai6) | JAX, #007906 | MGI:3809522 | Mouse line used in this study. |
| Genetic reagent (*Mus musculus*) | mTmG | JAX, #007676 | MGI:3716464 | Mouse line used in this study. |
| Genetic reagent (*Mus musculus*) | PolyloxExpress | Drs. H. Rodewald and A. Bhandoola | MGI:6470648 | Mouse line used in this study. |
| Genetic reagent (*Mus musculus*) | Runx1 Knock-In | Drs. Q. Ma and N. Speck | MGI:7490340 | Mouse line used in this study. |
| Sequence-based reagent | mTmG mouse strain genotyping primers (5′–3′) | CTT TAA GCC TGC CCA GAA GA TAG AGC T TG CGG AAC C CT TC AGG GA G CTG CAG TGG AGT AG | JAX: 007676 | |
| Sequence-based reagent | ZsGreen mouse strain genotyping primers (5′–3′) | AAG GGA GCT GCA GTG GAG TA CCG AAA ATC TGT GGG AAG T C GGC ATT AA A GCA GCG TA T CC AAC CAG AAG TGG CAC CTG AC | JAX: 007906 | |

*Appendix 1 Continued on next page*

*Appendix 1 Continued*

| Reagent type (species) or resource | Designation | Source or reference | Identifiers | Additional information |
|---|---|---|---|---|
| Sequence-based reagent | Cdh5-CreERT2(PAC) mouse strain genotyping primers (5'–3') | TCC TGA TGG TGC CTA TCC TC CCT GTT TTG C AC GTT CAC CG CAC CCT GTT CT T TGC CTC CT | This study | |
| Sequence-based reagent | Cdh5-CreERT2(BAC) mouse strain genotyping primers (5'–3') | ATA CCG GAG ATC ATG CAA GC ATG TGA A CC AGC TCC C TG TC CTA GG C CAC AGA AT T GAA AGA TC T GTA GGT GGA AAT TCT AGC AT C ATC C | JAX: Protocol 029211 | |
| Sequence-based reagent | Col1a2-CreERT mouse strain genotyping primers (5'–3') | CAT GTC CAT CAG GTT CTT GC TGA AAA A GT CCA CTA ATT AAA ACC A CTA ACA ACC CTT TCT CTC AAG GT CAG GAG GTT TCG ACT AAG TTG G | JAX: Protocol 19893 | |
| Sequence-based reagent | Runx1 Knock-In mouse strain genotyping primers (5'–3') | GAG TTC TCT GCT GCC TCC TGG CGA GGG CAG CCA TAG CAA CTC CGA GGC GGA TCA CAA GCA ATA | *Qi et al., 2017* | |
| Sequence-based reagent | PolyloxExpress mouse strain genotyping primers (5'–3') | AAG GGA GCT GCA GTG GAG TA TAA GCC TGC CCA GAA GAC T CC AAG ACC G CG AAG AGT T TG TCC | *Pei et al., 2020* | |
| Software, algorithm | FlowJo v10 | BD | https://www.flowjo.com RRID:SCR_008520 | |
| Software, algorithm | GraphPad Prism 10 | Dotmatics | www.graphpad.com RRID:SCR_002798 | |
| Software, algorithm | Snakemake | Köster et al. | https://snakemake.github.io/ RRID:SCR_003475; v9.6.0 | |
| Software, algorithm | Lima | PacBio | https://lima.how/ RRID:SCR_025520 | |
| Software, algorithm | Iso-Seq | PacBio | https://isoseq.how/ RRID:SCR_025481 | |
| Software, algorithm | Minimap2 | *Li, 2021* | https://github.com/lh3/minimap2 RRID:SCR_018550; Minimap2-2.28 (r1209) | |

*Appendix 1 Continued on next page*

*Appendix 1 Continued*

| Reagent type (species) or resource | Designation | Source or reference | Identifiers | Additional information |
|---|---|---|---|---|
| Software, algorithm | Cell Ranger | 10x Genomics | V8.0.1, RRID:SCR_017344 | |
| Software, algorithm | Polylox barcode recovery | This study; *Zhao, 2026* | https://github.com/CCRSF-IFX/SF_Polylox-BC | Custom or study-specific computational resource used in this study. |
| Software, algorithm | pGen calculation | *Pei et al., 2020*; *Rößler, 2017* | https://github.com/hoefer-lab/polylox | |
| Software, algorithm | Scanpy | Scanpy Community | https://scanpy.readthedocs.io RRID:SCR_018139 | |
| Software, algorithm | rapids-singlecell | Rapids-SingleCell | https://rapids-singlecell.readthedocs.io | |
| Software, algorithm | decoupler | *Badia-I-Mompel et al., 2022* | https://decoupler-py.readthedocs.io | |
| Software, algorithm | scANVI | *Gayoso et al., 2022* | https://github.com/scverse/scvi-tools; scvi-tools 1.4.0 | |
| Software, algorithm | CellTypist | *Xu et al., 2023* | https://celltypist.readthedocs.io/ | |
| Other | Primaria dishes/flasks | Corning | 353808; 353810; 353846 | Tissue culture surface used for primary endothelial cell culture without OP9 cells. |
| Other | Regular Cell Culture Flask | Corning | 430641U | |

