## [Editor Report · eLife Assessment]

This manuscript by Feng et al. provides **valuable** evidence regarding the hematopoietic differentiation of bone marrow endothelial cells in the adult mouse. Overall, the authors have addressed our main concerns. **Solid** data now more strongly support long-term multi-lineage reconstitution of the adult hemogenic endothelial cells. However, critical data, especially regarding the endothelial cells' hematopoietic identity and functional capacity, remain insufficient, which limits the strength of the hemogenic claim, especially the assertion that these adult hemogenic ECs generate bona fide HSCs. Additional experiments would be necessary to fully rule out alternative explanations.

---

## [Referee Report · Reviewer #1 (Public review)]

Summary:

This manuscript by Feng et al. uses mouse models to study the embryonic origins of HSPCs. Using multiple types of genetic lineage tracing, the authors aimed to identify whether BM-resident endothelial cells retain hematopoietic capacity in adult organisms. Through an important mix of various labeling methodologies (and various controls), they reach the conclusion that BM endothelial cells contribute up to 3-4% of hematopoietic cells in young mice.

Strengths:

The major strength of the paper lies in the combination of various labeling strategies, including multiple Cdh5-CreER transgenic lines, different CreER lines (col1a2), and different reporters (ZsGreen, mTmG), including a barcoding-type reporter (PolyLox). This makes it highly unlikely that the results are driven by a rare artifact due to one random Cre line, or one leaky reporter. The transplantation control (where the authors show no labeling of transplanted LSKs from the Cdh5 model) is also very supportive of their conclusions.

Weaknesses:

While the updated manuscript now provides strong evidence for Cdh5-CreER+ cells as a source of myeloid-biased hematopoiesis, the true identity of these "adult EHT stem cells", their differentiation hierarchy, the kinetics, the EHT mechanism, and the physiological relevance of this process remain unaddressed.

---

## [Referee Report · Reviewer #2 (Public review)]

Summary:

Feng, Jing-Xin et al. studied the hemogenic capacity of the endothelial cells in the adult mouse bone marrow. Using Cdh5-CreERT2 in vivo inducible system, though rare, they characterized a subset of endothelial cells expressing hematopoietic markers which was transplantable. They suggested that the endothelial cells need the support of stromal cells to acquire blood forming capacity ex vivo. This endothelial cells were transplantable and contribute to hematopoiesis with ca. 1% chimerism in a stress hematopoiesis condition (5-FU) and recruited to peritoneal cavity upon Thioglycolate treatment. Ultimately, the authors detailed the blood lineage generation of the adult endothelial cells in a single cell fashion suggesting a predominant HSPCs-independent blood formation by adult bone marrow endothelial cells, in addition to the discovery of Col1a2+ endothelial cells with blood forming potential corresponds to its high Runx1 expressing property.

Comments on revised version:

Overall, the authors have addressed our main concerns, and the revised manuscript is improved. The new data now more strongly supports long-term multilineage reconstitution of the adult hemogenic ECs. However, critical data, especially regarding the ECs' hematopoietic identity and functional capacity remains insufficient, which limits the strength of hemogenic claim, especially to assert that these adult hemogenic ECs generate bona fide HSCs.

Points that are sufficiently addressed:

(1) Exclusion of the potential contamination during cell sorting for the ex vivo CD45- ZsGreen+ fraction culture has been explicitly shown to be of a high purity in Fig. 2B.

(2). The pre-cultured ZsG+ fraction is shown to having a long-term multilineage reconstituting capacity (10 months chimerism, Fig. 2J), which increases confidence that the fraction is not limited to short-lived progenitors.

Points that are insufficiently addressed:

(1) As noted in the "Limitation of Study", the absence of LT-HSC phenotyping and/or secondary transplantation data of ZsG+ donor limits confidence in concluding that the adult hemogenic BM-ECs generate HSPCs.

(2) The lack of early to the end of reconstitution kinetics in Fig 2E-2J restricts interpretation of whether the donor fraction contains rapid reconstituting transient progenitor versus sustained repopulating HSCs.

---

## [Author Response]

The following is the authors’ response to the original reviews.

**Public Reviews:**

**Reviewer #1 (Public review):**
Summary:This manuscript by Feng et al. uses mouse models to study the embryonic origins of HSPCs. Using multiple types of genetic lineage tracing, the authors aimed to identify whether BM-resident endothelial cells retain hematopoietic capacity in adult organisms. Through an important mix of various labeling methodologies (and various controls), they reach the conclusion that BM endothelial cells contribute up to 3% of hematopoietic cells in young mice.Strengths:The major strength of the paper lies in the combination of various labeling strategies, including multiple Cdh5-CreER transgenic lines, different CreER lines (col1a2), and different reporters (ZsGreen, mTmG), including a barcoding-type reporter (PolyLox). This makes it highly unlikely that the results are driven by a rare artifact due to one random Cre line or one leaky reporter. The transplantation control (where the authors show no labeling of transplanted LSKs from the Cdh5 model) is also very supportive of their conclusions.

We appreciate the Reviewer’s consideration of the strengths of our study supporting the identification of adult endothelial to hematopoietic transition (EHT) in the mouse bone marrow.

Weaknesses:We believe that the work of ruling out alternative hypotheses, though initiated, was left incomplete. We specifically think that the authors need to properly consider whether there is specific, sparse labeling of HSPCs (in their native, non-transplant, model, in young animals). Polylox experiments, though an exciting addition, are also incomplete without additional controls. Some additional killer experiments are suggested.

Recognizing the importance of the weaknesses pointed by the Reviewer, we provide below our response to the thoughtful recommendations rendered.

**Reviewer #1 (Recommendations for the authors):**
The main model is to label cells using Cdh5 (VE-cadherin) CreERT2 genetic tracing. Cdh5 is a typical marker of endothelial cells. The data shows that, when treating adults with tamoxifen, the model labels PBMCs after ~10 days, and the labeling kinetics plateau by day 14... The authors reach the main conclusion: that adult ECs are making hematopoietic cells.

We agree that the main tool used in this study is to label endothelial cells (ECs) using Cdh5 (VE-Cadherin) CreERT2 genetic tracing in mice. Indeed, Cdh5 is recognized as a good marker of ECs. As a minor point, we wish to clarify that the results from treating adult Cdh5-CreERT2 mice with tamoxifen (Figure 1F) show that the ZsGreen labeling kinetics plateau by day 28 (not by day 14).

Important controls should be shown to rule out alternative possibilities: namely, that the CreERT2 reporter is being sparsely expressed in HSPCs. Many markers, specific as they may seem to be, can show expression in non-specific lineages - particularly in the cases of BAC and PAC transgenic models, in which the transgene can be present in multiple tandem copies and subject to genome location-specific effects. As the authors remind readers, the Cdh5 gene is partly transcribed (though at low levels) in HSPCs, and even more clearly expressed in specific subpopulations such as CLPs, DCs, pDCs, B cells, etc. Some options would be to: (i) check if the Cdh5-CreERT2 transgene (not endogenous Cdh5, but the BAC/PAC transgene) is expressed in LSKs (at least by qPCR), (ii) verify if any CreERT2 protein levels are present in LSKs (e.g., by western blot), and (iii) check if tamoxifen is labeling any HSPCs freshly after induction (e.g., flow cytometry data of ZsGreen LSKs at 24-48h post tamoxifen injection).

We fully agree with the Reviewer that many markers, allegedly specific to a certain cell type, can show expression in other cell lineages. We also agree that excluding sparse or ectopic CreERT2 expression in hematopoietic stem and progenitor cells (HSPCs) is essential for interpreting lineage-tracing results. As suggested by the Reviewer, we have now examined if the Cdh5-CreERT2 transgene is expressed in bone marrow LSKs. To this end, we analyzed the Polylox single-cell RNAseq dataset presented in this study, containing ZsGreen^+^ ECs and enriched ZsGreen^+^ LSKs. As shown in the revised Figure S4D, CreERT2 transcripts were detected exclusively in Cdh5-expressing endothelial populations and were absent from Ptprc/CD45-expressing hematopoietic cells, except for plasmacytoid dendritic cells (pDCs; Figure S4E). These results are consistent with the RNAseq data from adult mouse bone marrow[1] showing that the Cdh5 gene is not expressed in HSPCs, CLPs, DCs, or B cells. Rather, among hematopoietic CD45^+^ cells, Cdh5 is only expressed in a small subset of plasmacytoid dendritic cells (pDCs), which are terminally differentiated cells. These published results are described in the text.

To further support this conclusion, we provide additional single-cell RNAseq analyses from our unpublished dataset of LSKs isolated from Cdh5-CreERT2/ZsGreen mice and not enriched for ZsGreen expression. These new analyses were performed after integrating the single-cell data from ECs and ZsGreen^+^ hematopoietic cells from the Polylox dataset (current study). As shown in Author response images 1 and 2, CreERT2 expression closely matches the expression patterns of *Cdh5*, *Pecam1*, and *Emcn* and is not detected in Ptprc/CD45-expressing hematopoietic cells.

**Author response image 1. sa3fig1:** Expression of CreERT2, Cdh5, Ptprc and ZsGreen in BM cell populations enriched with ECs and hematopoietic cells. The single-cell RNAseq results are derived from ZsGreen-enriched BM ECs and ZsGreen-enriched BM hematopoietic cells were derived from Polylox lineage-tracing experiments (data shown in Fig. 5; 37,667 ECs and 48,065 BM hematopoietic cells) and from LSKs (23,017 cells) independently isolated from tamoxifen-treated Cdh5-CreERT2/ZsGreen mice without ZsGreen enrichment (unpublished data).

**Author response image 2. sa3fig2:** Expression of *CreERT2*, *Cdh5*, *Ptprc*, *Pecam1*, *Emcn*, *ZsGreen1*, *Col1a2*, *Cd19*, *Cd3e*, *Itgam* (*CD11b*), *Ly6a (Sca-1)*, *Kit*(*cKit*), *Cd34*, *Cd48*, *Slamf1* (CD150), and Siglech in enriched BM ECs and LSKs from Cdh5-CreERT2/ZsGreen mice treated with tamoxifen 4 weeks prior to harvest (same cell source as indicated in Author response image 1).

Additionally, we functionally tested whether hematopoietic progenitors could acquire ZsGreen labeling following tamoxifen administration using transplantation assays (Figure 4A-D). ZsGreen^-^ LSKs (purity 99%), sorted from Cdh5-CreERT2/ZsGreen donors that had never been exposed to tamoxifen to exclude background Cre leakiness, were transplanted into lethally irradiated wild-type recipients. After stable hematopoietic reconstitution, recipients were treated with tamoxifen. If transplanted HSPCs or their progeny expressed CreERT2, tamoxifen administration would be expected to induce ZsGreen labeling. However, no ZsGreen^+^ hematopoietic cells were detected in these recipients, demonstrating that hematopoietic progenitors from Cdh5-CreERT2/ZsGreen and their descendants do not undergo tamoxifen-induced recombination.

Together, the single-cell transcriptional and transplantation data demonstrate that CreERT2 expression and tamoxifen-induced recombination are restricted to Cdh5-expressing ECs (except for pDCs). These findings support the conclusion that ZsGreen^+^ hematopoietic cells arise from adult bone marrow ECs rather than from contaminating hematopoietic progenitors.

One important missing experiment is to trace how ECs actually do this hematopoietic conversion: meaning, which populations of HSPCs are being produced by adult ECs in the first instance? LT-HSCs? ST-HSCs? MPPs? GMPs? All of the above? What are the kinetics? Differentiation is likely to follow a hierarchical path, but this is unclear at the moment.

We agree that defining the earliest EC-derived hematopoietic cell progenitors and the kinetics by which these progenitors appear (LT-HSC vs ST-HSC/MPP vs lineage-restricted progenitors) would provide important insights into adult EHT.

In the current genetic labeling system, a rigorous kinetic analysis of hematopoietic cells first generated by EC-derived in vivo is not straightforward. Specifically, the low-level baseline reporter ZsGreen^+^ fluorescence in hematopoietic cells dependent on EHT occurring prenatally, perinatally or in young mice or other causes (Figure 1 A-D and Figure S1 D-I) impairs identification of newly generated ZsGreen^+^ progenitors at early time points and distinguish them from baseline fluorescence. A potential solution might be to introduce serial harvests across multiple time-points in large mouse cohorts to capture rare transitional events with statistical significance.

We wish to emphasize that the primary objective of this study was to establish whether adult bone marrow ECs have a hemogenic potential. Our data demonstrate adult EC-derived hematopoietic cell output that includes progenitor-containing fractions and multilineage mature progeny, under both steady-state conditions. We acknowledge that the current work does not resolve the order and kinetics of hematopoietic cell emergence following EHT. Therefore, under “Limitations of the study” we explicitly state this limitation and frame the identification of the earliest endothelial-derived progenitors and their kinetics as an important direction for future work.

One warning sign is how rare the reported phenomenon is. Even when labeling almost 90% of the BM ECs, these make at most ~3% of blood (less than 1% in the transplants in Figure 4F, less than 0.5% in the col1a2 tracing in Figure 7). This means this is a very rare and/or transient phenomenon... The most major warning sign is the fast kinetics of labeling and the fast plateau. We know that: (a) differentiation typically follows some hierarchy, (b) in situ dynamics of blood production are slow (work by Rodewald and Höfer). Considering how fast these populations need to be replaced to reach a steady state so rapidly (as reported here, 2-4 weeks), the presumably specialized ECs would need to be steadily dividing and producing hematopoietic cells at a fast pace (as a side prediction, the adult "EHT" cluster would likely be highly Mki67+). More importantly, the ZsGreen LSKs produced by the ECs would have to undergo VERY rapid differentiation (much faster than normal LSKs) or otherwise, if 3% of them are produced by a top compartment (the BM ECs) every 4 weeks, then the labeled population would continue to grow with time. The authors could try to challenge this by testing if the ZsGreen LSKs undergo much faster differentiation kinetics or lower self-renewal (which does not seem to be the case, at least in their own transplantation data). We believe a more likely explanation is that the label is being acquired more or less non-specifically, directly across a bunch of HSPC populations.

The Reviewer correctly notes that that the population of hemogenic ECs in the adult mouse bone marrow is small and the output of hematopoietic cells from these hemogenic ECs accounts for at most 3% of blood cells. We agree that delineating the kinetics by which hematopoietic cells are generated from adult EC is important, as this information would provide important insights into adult EHT.

Nonetheless, we believe that the rapid appearance and early plateau of labeled blood cells in our experiments may not derive from a sustained, high-rate generation of labeled blood cells from self-renewing top-tier hematopoietic cell compartments, such as LT-HSCs. Rather, our data are more consistent with a predominantly lineage-restricted and biased hematopoietic progenitor cell population being the source of labeled blood cells. Supporting this interpretation, longitudinal analysis of peripheral blood shows that EGFP^+^ PBMCs are consistently enriched with myeloid cells, whereas EGFP^-^ PBMCs are predominantly B cells (Figure 4G and H). This myeloid lineage skewing is stable over time and contrasts with what would be expected if labeling were acquired broadly and nonspecifically across the hematopoietic hierarchy. Therefore, our results are more consistent with myeloid biased progenitors being among the first populations that EHT generates.

We acknowledge that our studies do not identify the earliest endothelial-derived hematopoietic cells produced in vivo, and do not define their differentiation kinetics. Addressing rigorously these questions would require temporally resolved lineage tracing with sufficiently powered cohorts at early time point to statistically distinguish from baseline reporter background. These important experiments were beyond the scope of the present study. As noted above, under “Limitations of the study” we explicitly state this limitation and frame the identification of the earliest endothelial-derived progenitors and their kinetics as an important direction for future work.

Transplant experiments in Figure 4 do offer a crucial experiment in support of the main conclusion of the manuscript. These experiments show that transplanted LSKs bearing the Cdh5-CreERT2 and ZsGreen reporter cannot acquire the tamoxifen-induced label post-transplantation - suggesting that the label is coming from ECs. However, it is also possible that the LSK Cdh5-CreERT expression is partly during the transplantation process... Indeed, we know through the aging data that the labeling is less active in aged mice. In any case, this would be verified by qPCR/western-blot (comparing native vs post-transplant LSKs).

We agree with the Reviewer that the experiment in Figure 4A-D “offer a crucial experiment in support of the main conclusion of the manuscript.” The results of this experiment show that ZsGreen negative LSKs from the Cdh5-CreERT2-ZsGreen reporter mice do not acquire tamoxifen-induced ZsGreen fluorescence post transplantation, supporting the endothelial cell origin of blood ZsGreen^+^ cells.

The Reviewer raises the possibility a “that the LSK Cdh5-CreERT expression is partly during the transplantation process... , and that this Cdh5-CreERT expression may occur slowly as learned “through the aging data that the labeling is less active in aged mice.” As we show in Figure 3F, tamoxifen administration induced a similar percentage of ZsGreen^+^ ECs in the bone marrow of Cdh5-Cre^ERT2^(BAC)/ZsGreen mice, whether tamoxifen was administered to 6-week-old, 16-week-old, 26-week-old or 36-week-old mice. Similar results with Cdh5-CreERT2 (BAC) mice are reported in the literature[2]. Since the mice transplanted with ZsGreen^-^ LSKs were followed for 25 weeks after tamoxifen administration, we believe that the results in Figure 4A-D address the concern raised by the Reviewer.

Supporting the conclusion that LSKs from the Cdh5-CreERT2-ZsGreen reporter mice do not express the Cdh5-CreERT2 under a native -non-transplant- setting, we now provide transcriptomic data from Cdh5-CreERT2/ZsGreen mice (not transplanted) showing that CreERT2 expression closely tracks with expression of canonical endothelial markers (Cdh5, Pecam1, Emcn) and is not detectable in Ptprc/CD45-expressing hematopoietic cells (Author response images 1 and 2). These data were obtained from non-transplanted mice treated with tamoxifen at ~12 weeks of age and analyzed four weeks later. Together, these results indicate that CreERT2 expression is endothelial-restricted in Cdh5-CreERT2-ZsGreen reporter mice.

Figure 5 presents PolyLox experiments to challenge whether adult ECs produce hematopoietic cells through in situ barcoding. Several important details of the experiment are missing in the main text (how many cells were labeled, at which time point, how long after induction were the cells sampled, how many bones/BM-cells were used for the sample preparation, what was the sampling rate per population after sorting, how many total barcodes were detected per population, how many were discarded/kept, what was the clone-size/abundance per compartment). As presented, the authors imply that 31 out of ~200 EC barcodes are shared with hematopoietic cells... This would suggest that ~15% of endothelial cells are producing hematopoietic cells at steady state. This does not align well with the rarity of the behavior and the steady state kinetics unless any BM EC could stochastically produce hematopoietic cells every couple of weeks, or if the clonality of the BM EC compartment would be drastically reduced during the pulse-chase overlap with mesenchymal cells. Important controls are missing, such as what would be the overlap with a population that is known to be phylogenetically unrelated (e.g., how many of these barcodes would be found by random chance at this same Pgen cut-off in a second induced mouse). Also, the Pgen value could be plotted directly to see whether the clones with more overlapping populations/cells (3HG, 127, 125, CBA) also have a higher Pgen. We posit that there are large numbers of hematopoietic clones that contribute to adult hematopoiesis (anywhere from 2,000-20,000 clones would be producing granulocytes after 16 weeks post chase), and it would be easy to find clones that overlap with granulocytes (the most abundant and easily sampled population) - HSPCs would be the more stringent metric.

We thank the Reviewer for highlighting the need for a more detailed description of the Polylox experiments. To address this deficiency, we have compiled a document (Additional Supplementary Information file) containing all the specifics of the Polylox experimental and analytical parameters in one location. This includes: (i) the number of cells analyzed per population, (ii) the time points of induction and sample collection, (iii) the number of bones and total bone marrow cells used for preparation, (iv) the sampling rate following cell sorting, (v) the total number of detected barcodes per population, (vi) barcode filtering criteria and numbers retained or discarded, and (vii) clone-size and barcode number across cell compartments. We have updated the manuscript to refer readers to this Supplementary file.

The Reviewer concluded from our results (Figure 5, Figure S5) that 31 out of ~200 endothelial cell (EC) barcodes shared with hematopoietic cells (HCs), implying that ~15% of ECs produce hematopoietic cell progeny at steady state. This interpretation in inconsistent with our data showing the rare nature of adult EHT and would require either that a large fraction of bone-marrow ECs can generate hematopoietic cells within short time windows, or that EC would clonally expand rapidly during the pulse-chase period, as noted by the Reviewer. The explanation for this apparent problem is technical. Briefly, the ~200 EC barcodes recovered do not represent all barcoded ECs. During Polylox barcode library construction, a mandatory size-selection step is applied prior to PacBio sequencing, retaining fragments that are approximately 800–1500 bp in length, whereas the full Polylox cassette spans ~2800 bp. This is mainly because the PacBio sequencer requires that the library be either 800-1500bp or over 2500bp, for optimal sequencing results. As described in the original Polylox publication[3,4], this size selection eliminates most (approximately 75%) longer barcodes, together with ~85% of the shorter barcodes. Thus, ECs harboring very long or short recombined barcodes are under-represented or excluded from sequencing. As a result, the 22 true barcodes linking ECs and HCs recovered from sequencing do not indicate that ~10–15% of ECs generate hematopoietic progeny. Rather, these barcodes represent a highly selected subset of ECs with barcode configurations compatible with library recovery and sequencing. The observed EC–HC barcode sharing thus reflects qualitative lineage connectivity, not the quantitative frequency of endothelial-derived hematopoiesis at steady state.

The Reviewer correctly notes that true Polylox barcodes are shared by ECs and mesenchymal-type cells and asks that we examine whether this overlap could occur by chance alone. The Polylox filtering threshold (pGen < 1 × 10^-6^), that we have revised for stringency (from pGen < 1 × 10^-4^, without altering the essential results; new Figure S4 and revised Figure 5C-F) renders such overlap exceedingly unlikely. At this threshold, the expected number of random recombination events among 4,069 barcoded cells is approximately 0.004. Consequently, among the 87 mesenchymal cells identified here, fewer than 0.4 cells would be expected, to share a barcode with another cell by chance alone. Thus, the probability of recovering identical barcodes across unrelated lineages due to random recombination is vanishingly small, and the observed EC–mesenchymal barcode sharing substantially exceeds random expectation.

Related to this observation, the Reviewer correctly notes that the endothelial and mesenchymal cell lineages are phylogenetically unrelated. However, endothelial-to-mesenchymal cell transition (EndMT), the process by which normal ECs completely or partially lose their endothelial identity and acquire expression of mesenchymal markers, is a well-established process that occurs physiologically and in disease states (Simons M Curr Opin Physiol 2023). In the bone marrow, the occurrence of EndMT has been documented in patients with myelofibrosis, and the process affects the bone marrow microvasculature (Erba BG et al The Amer J Patholl 2017). Single-cell RNAseq of non-hematopoietic bone marrow cells has shown the existence of a rare population of ECs that co-expresses endothelial cell markers (Cdh5, Kdr, Emcm and others) and the mesenchymal cell markers, as shown in Figure 6E and F.

We fully agree with the Reviewer that given the large number of hematopoietic clones contributing to adult hematopoiesis -particularly granulocyte-producing clones- it may be relatively easy to detect barcode overlap with abundant mature populations, whereas overlap with HSPCs would represent a more stringent and informative metric of lineage relationships. The Polylox results presented here show the sharing of true barcodes between individual ECs and HSPC.

**Reviewer #2 (Public review):**
Summary:Feng, Jing-Xin et al. studied the hemogenic capacity of the endothelial cells in the adult mouse bone marrow. Using Cdh5-CreERT2 in vivo inducible system, though rare, they characterized a subset of endothelial cells expressing hematopoietic markers that were transplantable. They suggested that the endothelial cells need the support of stromal cells to acquire blood-forming capacity ex vivo. These endothelial cells were transplantable and contributed to hematopoiesis with ca. 1% chimerism in a stress hematopoiesis condition (5-FU) and recruited to the peritoneal cavity upon Thioglycolate treatment. Ultimately, the authors detailed the blood lineage generation of the adult endothelial cells in a single cell fashion, suggesting a predominant HSPCs-independent blood formation by adult bone marrow endothelial cells, in addition to the discovery of Col1a2+ endothelial cells with blood-forming potential, corresponding to their high Runx1 expressing property.The conclusion regarding the characterization of hematopoietic-related endothelial cells in adult bone marrow is well supported by data. However, the paper would be more convincing, if the function of the endothelial cells were characterized more rigorously.

We thank the Reviewer for the supportive comments about our study.

(1) Ex vivo culture of CD45-VE-Cadherin+ZsGreen EC cells generated CD45+ZsGreen+ hematopoietic cells. However, given that FACS sorting can never achieve 100% purity, there is a concern that hematopoietic cells might arise from the ones that got contaminated into the culture at the time of sorting. The sorting purity and time course analysis of ex vivo culture should be shown to exclude the possibility.

We agree that FACS sorting can never achieve 100% cell purity and that sorting purity is critical for interpreting the ex vivo culture experiments presented in our study. As requested by the Reviewer, we have now documented the purity of the sorted endothelial cell (EC) population used in the ex vivo culture experiments. The post-sort purity of CD45/sup>VE-cadherin^+^ZsGreen^+^ ECs was 96.5 %; this data is now shown in the revised Figure 2B (Post Sort Purity panel). This purity level is comparable to purity levels of sorted ECs shown in Figure S2I (94.5 %).

While we agree that a detailed time-course analysis of hematopoietic cell output from EC cultures could further strengthen the conclusion that bone marrow ECs can produce hematopoietic cells ex vivo, we wish to call attention to the additional critical control in the experiment shown in Figure 2B-D. In this experiment, we co-cultured CD45^+^ZsGreen^+^ hematopoietic cells from Cdh5-CreERT2/ZsGreen mice, rather than ECs, and examined if these hematopoietic cells could produce ZsGreen^+^ cell progeny after 8-week culture under the same conditions used in EC co-cultures (conditions not designed to support hematopoietic cells long-term). Unlike ECs, the CD45^+^ZsGreen^+^ hematopoietic cells did not generate ZsGreen^+^ hematopoietic cells at the end of the 8-week culture, indicating that the culture conditions are not permissive for the maintenance, proliferation and differentiation of hematopoietic cells. This provides strong evidence that even if few hematopoietic cells contaminated the sorted ECs, these hematopoietic cells would not contribute to EC-derived production of hematopoietic cells at the 8-week time-point. We have revised the text of the results describing the results of Figure 2B-D.

(2) Although it was mentioned in the text that the experimental mice survived up to 12 weeks after lethal irradiation and transplantation, the time-course kinetics of donor cell repopulation (>12 weeks) would add a precise and convincing evaluation. This would be absolutely needed as the chimerism kinetics can allow us to guess what repopulation they were (HSC versus progenitors). Moreover, data on either bone marrow chimerism assessing phenotypic LT-HSC and/or secondary transplantation would dramatically strengthen the manuscript.

The original manuscript reported survival and engraftment up to 12 weeks post transplantation. The recipient mice have now been monitored for up to 10 months post transplantation. These extended survival and engraftment data are now included in the revised Figure 2I and J replacing the previous 10-week analyses.

We agree with the Reviewer that the time-course kinetics of donor cell repopulation would help define adult endothelial to hematopoietic transition (EHT) and the hematopoietic cell types produced by adult (EHT). We did not perform serial time-course sampling of peripheral blood beyond the 10-week and the 10-month time-points. Given that the recipient mice were lethally irradiated with increased susceptibility to infection, we sought to minimize repeated interventions that could compromise animal health and survival. We therefore prioritized long-term survival and endpoint analysis over repeated longitudinal sampling. Nonetheless, the long-term survival,10 months, and multilineage hematopoietic cell reconstitution after lethal irradiation provides functional evidence that adult EHT produced at least some LT-HSC.

We acknowledge that phenotypic assessment of bone marrow LT-HSC chimerism /or secondary transplantation would further strengthen the manuscript. We have clarified these limitations in the revised manuscript under “Limitations of the study”.

(3) The conclusion by the authors, which says "Adult EHT is independent of pre-existing hematopoietic cell progenitors", is not fully supported by the experimental evidence provided (Figure 4 and Figure S3). More recipients with ZsGreen+ LSK must be tested.

We agree with the Reviewer that, in most cases, a larger number of experimental data points is helpful to strengthen the conclusions, and that having additional mice transplanted with ZsGreen-enriched LSK would be desirable. However, we do not believe that additional mice transplanted with ZsGreen LSKs would strengthen the conclusions drawn from the experimental results shown in Figure 4D, in which we used 6 mice transplanted with ZsGreen-depleted (ZsGreen^-^) LSKs and 2 mice transplanted with ZsGreen^+^-enriched (ZsGreen^+^) LSKs. The independence of adult EHT from “pre-existing hematopoietic cell progenitors” is based on the following experimental results and conclusion from these results.

First, ZsGreen^-^ LSKs (purity 99%) isolated from Cdh5-CreERT2/ZsGreen mice were transplanted into lethally irradiated WT recipients (n = 6). These ZsGreen^-^ LSKs robustly reconstituted hematopoiesis, demonstrating successful engraftment. Importantly, tamoxifen administration to the recipients of ZsGreen^-^ LSKs produced no detectable ZsGreen^+^ cells in the blood for up to 6 months post transplantation (Figure 4D, blue line encompassing the results of the 6 mice). This result demonstrates that the transplanted ZsGreen^-^ hematopoietic progenitors and their progeny do not acquire ZsGreen labeling in vivo following tamoxifen treatment, indicating that they lack the Cre-recombinase. This result is consistent with the endothelial specificity of Cdh5 expression.

Second, ZsGreen^+^ LSKs (accounting for ~50% of the LSKs) isolated from Cdh5-CreERT2/ZsGreen mice were transplanted into lethally irradiated WT recipients (n = 2). This arm of the experiment was performed in part as a technical control to confirm successful engraftment and detection of ZsGreen^+^ hematopoietic cells in the transplant setting. Importantly, tamoxifen administration to the two recipients of ZsGreen^+^ LSKs (Figure 4D, two green lines reflecting these two mice) show that the level of ZsGreen^+^ blood cells stabilized in each of the mice between week 10 and 24, showing equilibrium between the proportion of ZsGreen^+^ and ZsGreen^-^cells in the blood. This indicates that pre-existing ZsGreen^+^ LSK are not responsible for tamoxifen-induced increases in ZsGreen^+^ hematopoietic cell in blood.

Together, the results from this experiment demonstrate that in the setting of transplantation, tamoxifen does not induce ZsGreen labeling of ZsGreen- hematopoietic progenitors/their progeny. This result strongly supports the conclusion that ZsGreen⁺ hematopoietic cells arise independently of pre-existing or inducible hematopoietic progenitors. We have revised the text to clarify these experiments and to present the results in a simplified manner.

Strengths:The authors used multiple methods to characterize the blood-forming capacity of the genetically - and phenotypically - defined endothelial cells from several reporter mouse systems. The polylox barcoding method to trace the adult bone marrow endothelial cell contribution to hematopoiesis is a strong insight to estimate the lineage contribution.Weaknesses:It is unclear what the biological significance of the blood cells de novo generated from the adult bone marrow endothelial cells is. Moreover, since the frequency is very rare (<1% bone marrow and peripheral blood CD45+), more data regarding its identity (function, morphology, and markers) are needed to clearly exclude the possibility of contamination/mosaicism of the reporter mice system used.

We agree that the biological significance and functional roles of hematopoietic cells generated de novo from adult bone marrow ECs remain important open questions. We also agree that the output of hematopoietic cells from adult EHT is low, but rare events can be important, particularly as they pertain to stem/progenitor cell biology. Both points are described under “Limitations of the study”. The primary goal of the present study was to address the question whether adult bone marrow ECs can undergo EHT. We believe that the combination of various mouse transgenic lines, different Cre-ER, different reporters (ZsGreen and mTmG), including the s.c. barcoding reporter (PolyloxExpress), different approaches to evaluate hematopoiesis in vivo and ex vivo, makes it rather unlikely that our conclusions are driven by an artifact related to a specific leaky reporter, contamination, or problems with one of the Cre-lines. The experiment where we find no tamoxifen-induced labeling of transplanted ZsGreen^-^ LSKs derived from the Cdh5-CreERT2/ZsGreen mice is strongly supportive of the existence of adult EHT, virtually excluding a contribution of contaminant hematopoietic cells.

**Reviewer 2 Recommendations for the authors:**
(1) There is a discrepancy in the proportion of peripheral blood composition between different reporters (mTmG and ZsGreen) (Figure 1G and Figure S1K), especially the contrasting B cell proportion between both models. The additional comments on this data should be mentioned.

In the revised Results section, we now note that the mTmG and ZsGreen reporters show slightly different efficiencies or kinetics of labeling. These differences have previously been reported[5] and have been attributed to relative reporter leakiness, sensitivity to tamoxifen, or different kinetics of Cre recombination. As suggested, these comments have been added to the text following the description of (Figure S2A).

(2) Experimental methods concerning cell transplantation/transfer need more information, such as: (a) using or not using rescue cells and how many cells are they if using, (b) single or split dose of irradiation, (c) when were cells transplanted following irradiation, etc. Otherwise, the data are uninterpretable.

We have ensured that the Material and Methods section under “Bone marrow ablation and transplantation” contains all the information requested by the Reviewer.

(3) Some of the grouped data haven't been statistically analyzed.

We have reviewed all data and performed appropriate statistical analyses where comparisons were made. In the revised figures and legends, all grouped datasets now include statistical tests and p-values are indicated (added to Fig. 3H and I; Figure 4G).

(4) Some flowcytometry plot has the quantitative number, others do not. The quantitative information is absolutely needed in all flow cytometry plots.

We have updated the flow cytometry figures to include quantitative values (percentages or absolute counts) in all relevant plots (2B (new figure, bottom left); 2C; S1G, S1H).

(5) It is more relevant to present the Emcn/VE-Cadherin plot from gated CD45+/ZsGreen+, not the CD45-/ZsGreen+ fraction (Figure 2C), as the latter were not the EHT-derived offspring, but rather the common phenotypic endothelial cells

As requested, we have added the suggested flow cytometry plot. The revised Figure 2C now includes an Emcn vs. VE-Cadherin plot from the gated CD45^+^ZsGreen^+^ population. This complements the existing panel and confirms that the cells of interest retain endothelial cell markers after culture, while the CD45^+^ZsGreen^+^ cells did not express endothelial markers. The figure legend has been updated to explain the new panel. We agree that this plot more directly highlights the phenotype of the presumed EHT-derived cells.

(6) To show the effect of the ex vivo culture, the authors should present the absolute number of CD45+ZsGreen+ cells in the pre-/post-culture; otherwise, the data are uninterpretable (Figure 2D).

Our interpretation of the Reviewer’s comment above (relative to the experiment shown in Figure 2B-D) is that the Reviewer would like that we provide the absolute number of CD45^+^ZsGreen^+^ cells introduced into the co-culture (supplemented with unsorted BM cells, ZsGreen^+^ hematopoietic cell or ZsGreen^+^ ECs) and the absolute number of CD45^+^ZsGreen^+^ cells recovered at the end of the 8-week culture. Currently, the results in Figure 2D show the absolute number of CD45^+^ZsGreen^+^ cells recovered at the end of the 8-week culture. The input of CD45^+^ZsGreen^+^ cells for unsorted BM cells was 2.93e6 on average; for ZsGreen^+^ hematopoietic cells was 1.68e6 on average and from sorted ZsGreen^+^ ECs was estimate up to 100.

(7) It is confusing to see Figures 2F and 2G, which apparently show the data from the middle of the experimental procedure (Figure 2E). Those data should be labelled clearly regarding which procedures of the whole experiment protocol.

As correctly noted by the Reviewer, Figures 2F and 2G provide data that relate to the middle of the graphical representation of the experiment shown in Figure 2E. We see how this may be confusing.

Therefore, we have updated both the figure labeling and legend to explicitly indicate that Figure 2F and 2G provide the FACS sorting results for the cells used for transplantation. The revised legend now reads: “Representative flow cytometry plots of the non-adherent cell fraction after 8 weeks of co-culture (cells used for transplantation).”

References

(1) Kucinski, I., Campos, J., Barile, M., Severi, F., Bohin, N., Moreira, P.N., Allen, L., Lawson, H., Haltalli, M.L.R., Kinston, S.J., et al. (2024). A time- and single-cell-resolved model of murine bone marrow hematopoiesis. Cell Stem Cell 31, 244-259.e10. https://doi.org/10.1016/j.stem.2023.12.001.

(2) Identification of a clonally expanding haematopoietic compartment in bone marrow | The EMBO Journal | Springer Nature Link https://link.springer.com/article/10.1038/emboj.2012.308.

(3) Pei, W., Shang, F., Wang, X., Fanti, A.-K., Greco, A., Busch, K., Klapproth, K., Zhang, Q., Quedenau, C., Sauer, S., et al. (2020). Resolving Fates and Single-Cell Transcriptomes of Hematopoietic Stem Cell Clones by PolyloxExpress Barcoding. Cell Stem Cell 27, 383-395.e8. https://doi.org/10.1016/j.stem.2020.07.018.

(4) Pei, W., Feyerabend, T.B., Rössler, J., Wang, X., Postrach, D., Busch, K., Rode, I., Klapproth, K., Dietlein, N., Quedenau, C., et al. (2017). Polylox barcoding reveals haematopoietic stem cell fates realized in vivo. Nature 548, 456–460. https://doi.org/10.1038/nature23653.

(5) Álvarez-Aznar, A., Martínez-Corral, I., Daubel, N., Betsholtz, C., Mäkinen, T., and Gaengel, K. (2020). Tamoxifen-independent recombination of reporter genes limits lineage tracing and mosaic analysis using CreERT2 lines. Transgenic Res 29, 53–68. https://doi.org/10.1007/s11248-019-00177-8.